# A Unified Approach to Routing and Cascading for LLMs

Jasper Dekoninck[1]   Maximilian Baader[1]   Martin Vechev[1]

## Abstract

The availability of a wide range of large language models (LLMs) embedded in various agentic systems has significantly increased the potential of model selection strategies to improve the cost-performance tradeoff. Existing strategies involve either routing, where a single model is chosen per query, or cascading, which sequentially runs increasingly larger models until a satisfactory answer is found. However, current approaches face three key limitations: they (1) lack formal proofs of optimality, (2) fail to identify the conditions under which these strategies are most effective to improve the cost-performance tradeoff, and (3) are unable to combine both paradigms for further improvements. To address these issues, we first derive a novel optimal strategy for cascading and prove the optimality of an existing routing strategy. Further, we propose *cascade routing*, a unified framework that integrates routing and cascading into a theoretically optimal strategy. Through our analysis, we identify good quality estimators as the critical factor for the success of model selection paradigms. Finally, in our experiments, we show that cascade routing consistently outperforms the individual approaches by a large margin and we analyze quality estimators to determine when routing and/or cascading are useful paradigms for model selection.[1]

## 1. Introduction

Large language models (LLMs) have found applications in a wide range of tasks, some of which are easily handled by small models, while others require the full capacity of state-of-the-art LLMs. This has led to the development of many fine-tuned models of various sizes that target specific tasks.

To maximize performance, it is crucial to select the most suitable model for each query, accounting for both the expected quality of the model's output and the model's cost. Such model selection strategies can significantly improve performance over any individual model and can reduce inference costs by selecting a smaller model when the query does not require the full capacity of a larger model.

**Routing and Cascading**   Two primary strategies have been proposed to solve model selection. The first, routing, directs each input query to a specific model from a set of available models (Chen et al., 2022; Liu et al., 2024), as illustrated in Fig. 1(a). This approach is particularly effective when different expert LLMs are needed for diverse tasks, enabling the selection of the most suitable expert for each query. The second strategy, cascading, processes an input query through a sequence of increasingly larger models, stopping when a model produces an answer deemed sufficiently good (Chen et al., 2023; Varshney & Baral, 2022), as illustrated in Fig. 1(b). Cascading is particularly valuable for handling queries of varying difficulty, as it allows simpler queries to be addressed by smaller models while reserving more complex queries for larger models.

**Restrictive Conditions**   Despite their utility, both routing and cascading impose significant restrictions on the model selection process. In routing, the initial selection of a model is final, preventing any reconsideration after the initial decision. In cascading, each query must sequentially pass through all models in the chain, with no option to skip a model. Therefore, a less restrictive strategy that combines the strengths of both routing and cascading could offer significant performance improvements.

**Lack of Deeper Understanding**   Further, the conditions under which current routing and cascading strategies are optimal, are not well understood. For routing, an extensive proof is required just to show that current strategies are *close to* optimal (Chen et al., 2022), while the theoretical analysis of cascading does not provide optimality guarantees (Chen et al., 2023; Varshney & Baral, 2022). This lack of theoretical understanding hinders the development of more effective model selection strategies. Moreover, prior work fails to provide insights into the limitations of model selection strategies and cannot identify the conditions under which they are useful in practical scenarios.

---

[1]Department of Computer Science, ETH Zurich, Switzerland. Correspondence to: Jasper Dekoninck <jasper.dekoninck@inf.ethz.ch>.

*Proceedings of the 42nd International Conference on Machine Learning*, Vancouver, Canada. PMLR 267, 2025. Copyright 2025 by the author(s).

[1]Code available at `https://github.com/eth-sri/cascade-routing`

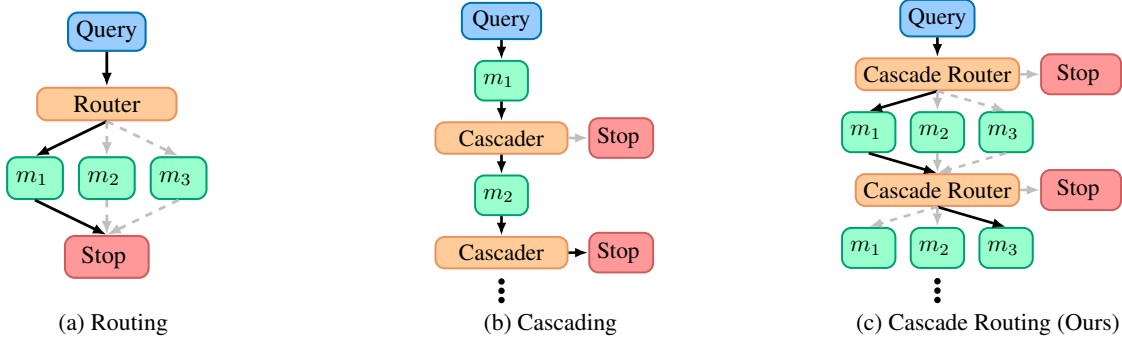

Figure 1: Overview of three model selection strategies. Routing selects a single model for a query, cascading processes queries through a sequence of models, and cascade routing generalizes both.

For instance, it is widely believed that one needs models of various sizes to benefit from cascading (Chen et al., 2023; Gupta et al., 2024; Khalili et al., 2022), but we show that this notion is incorrect.

**This Work: Cascade Routing**  To address these limitations, we first derive optimal routing and cascading strategies by framing them as linear optimization problems aimed at maximizing output quality while remaining within a given cost budget. For routing, this optimal strategy is close to the one obtained by prior work, while for cascading we derive a new strategy that is provably better than existing approaches. Building on this analysis, we propose a new paradigm called cascade routing, which generalizes both routing and cascading. As illustrated in Fig. 1(c), cascade routing initially routes a query to any available model but keeps rerouting to different models until a model produces an answer of sufficient quality. We prove the optimality of cascade routing and show that it offers significantly more flexibility in processing a query.

**Importance of Quality Estimation**  Our theoretical analysis enables a more thorough investigation into the factors that influence the effectiveness of model selection strategies. Specifically, we find that accurate estimates of model performance and response quality are most important. For routing, reliable *ex-ante quality estimation*—the ability to predict whether a model will perform well on a given query—is essential. For cascading, robust *post-hoc quality estimation*—the ability to evaluate the quality of a model's response after generation—is critical. Without it, the effectiveness of cascading strategies is severely limited even when models of various sizes are available.

**Results**  We evaluate cascade routing on a range of tasks, demonstrating that it significantly outperforms both routing and cascading. Notably, cascade routing consistently outperforms other methods, improving performance by up to $8\%$ on the RouterBench benchmark (Hu et al., 2024) and by $14\%$ on the SWE-Bench benchmark (Jimenez et al., 2024). Further, we show that our new cascading strategy outperforms existing cascades in several scenarios by over $10\%$.

**Key Contributions**  Our main contributions are:

- We derive optimal strategies for routing and cascading and obtain a new cascading strategy that is provably better than prior approaches (§2, §3).

- We introduce cascade routing, a new paradigm that combines the strengths of routing and cascading, and prove its optimality (§4).

- We conduct a thorough evaluation, demonstrating that cascade routing consistently outperforms the baselines, highlighting the critical role of quality estimation for the effectiveness of model selection (§5).

## 2. Routing as Linear Optimization

We derive an optimal routing strategy to select the best model for a given query, providing detailed proofs for all statements in this section in App. A.

**Brief Overview**  In this section, we begin by defining a routing strategy as a function that maps queries to models. Next, we introduce the notation for quality and cost functions, demonstrating how the optimal routing strategy can be formulated to maximize a linear tradeoff between these two factors. Lastly, we give an illustrative example and discuss the significance of quality estimation in routing and its impact on the effectiveness of the strategy.

**Routing**  In routing, our goal is to develop a strategy that selects the best language model for a given input query. Formally, let $\mathcal{X}$ represent the distribution over all possible queries, and suppose we have $k$ language models $m_1, \ldots, m_k$ available for routing. Further, let $\Delta_k$ denote the set of all probability distributions over $k$ variables. A routing strategy can then be defined as follows:

**Definition 1** (Routing). *A routing strategy $s$ is a function $s \colon \mathcal{X} \to \Delta_k$ that maps a query $x \in \mathcal{X}$ to a probability distribution over models. $s_i(x)$ denotes the probability that $m_i$ is selected for query $x$.*

A routing strategy selects a model by sampling from the distribution $s(x)$ for each query $x$. In prior work, routing strategies were restricted to be deterministic, i.e., $s_i(x) \in \{0, 1\}$ (Chen et al., 2022; Hu et al., 2024). In contrast, we propose using a more general probabilistic strategy that enables a better solution and an easier theoretical analysis.

**Quality and Cost** In routing, we seek to maximize the output quality of the selected model while adhering to a given cost budget $B$. We define the quality function $q_i(x)$ as the output quality of model $m_i$ on query $x$, and the cost function $c_i(x)$ as the cost of running model $m_i$ on $x$. Quality could measure model accuracy, user preference, or any other performance indicator. Cost could measure either monetary costs or latency, depending on the use case.

However, since these functions are unknown in practice, we need estimators $\hat{q}_i(x)$ and $\hat{c}_i(x)$ that approximate the output quality and cost of querying model $m_i$ on input $x$. Estimators for $q_i(x)$ can be created using small classifiers trained to predict model accuracy, as done in prior work (Hu et al., 2024; Shnitzer et al., 2023). $\hat{c}_i(x)$ can be estimated by tokenizing the input query and determining the average output length of the model on a query. Then, we can use API-specific costs per token to estimate the cost of running a model on a query. Alternatively, we can also use average execution time as a proxy for cost.

**Optimal Routing** Using these estimators, we can formally define the optimal routing strategy:

**Definition 2** (Optimal Routing)**.** *The optimal routing strategy $s_{\text{OPT}}$ for a given cost budget $B$ is the solution to the optimization problem that maximizes the expected output quality of the selected model while adhering to the budget:*

$$\max_{s} \quad \mathbb{E}_{x \in \mathcal{X}} \left( \sum_{i=1}^{k} s_i(x)\hat{q}_i(x) \right)$$
$$\text{s.t.} \quad \mathbb{E}_{x \in \mathcal{X}} \left( \sum_{i=1}^{k} s_i(x)\hat{c}_i(x) \right) \leqslant B. \tag{1}$$

We now explain how to solve this optimization problem. For a given query $x$, it can be shown (see App. A) that the optimal routing strategy selects the model maximizing the cost-quality tradeoff $\tau_i(x, \lambda) = \hat{q}_i(x) - \lambda\hat{c}_i(x)$. Here, $\lambda \in \mathbb{R}^+$ is a hyperparameter that controls the balance between quality and cost based on the budget $B$.

However, it can occur that several models achieve the same optimal cost-quality tradeoff for a given query. To address this, we define two deterministic strategies $s_{\text{MIN}}^{\lambda}(x)$ and $s_{\text{MAX}}^{\lambda}(x)$, which, respectively, select the cheapest and most expensive model that achieves the optimal tradeoff. The optimal routing strategy $s_{\text{OPT}}$ is then determined by:

**Theorem 1** (Optimal Routing Strategy)**.** *For a cost budget $B$, there exists a $\lambda \in \mathbb{R}^+$ and a $\gamma \in [0, 1]$ such that the optimal routing strategy $s_{\text{OPT}}$ equals $\gamma s_{\text{MIN}}^{\lambda} + (1 - \gamma)s_{\text{MAX}}^{\lambda}$.*

**Theorem 1**, continued. *Furthermore, all routing strategies that have an expected cost that is exactly equal to $B$ and can be written as a convex combination of $s_{\text{MIN}}^{\lambda'}$ and $s_{\text{MAX}}^{\lambda'}$ for some $\lambda' \in \mathbb{R}^+$ achieve the same optimal quality.*

In App. A, we show how to obtain the optimal $\lambda$ and $\gamma$ for a cost budget $B$ using a validation dataset $D$. In Algorithm 1, we provide pseudocode for the optimal routing algorithm.

---

**Algorithm 1** Optimal Routing Algorithm

**Input**: input query $x$, quality estimator $\hat{q}_i$, cost estimator $\hat{c}_i$, tradeoff parameters $\lambda$ and $\gamma$
**Output**: Model index $i$ to be used for query $x$

1: $\tau_i(x, \lambda) := \hat{q}_i(x) - \lambda\hat{c}_i(x)$
2: $\tau_{\max}(x, \lambda) := \max_{i \in \{1, \dots, k\}} \tau_i(x, \lambda)$
3: best $:= \{i \in \{1, \dots, k\} | \tau_i(x, \lambda) = \tau_{\max}(x, \lambda)\}$
4: min_cost_index $:= \arg\min_{i \in \text{best}} c_i(x)$
5: max_cost_index $:= \arg\max_{i \in \text{best}} c_i(x)$
6: **if** random$(0, 1) < \gamma$ **then**
7:     **return** min_cost_index
8: **else**
9:     **return** max_cost_index
10: **end if**

---

Since $\gamma$ is often not equal to $0$ or $1$, $s_{\text{OPT}}$ is not necessarily deterministic, i.e., there are queries $x$ such that $s_{\text{OPT},i}(x) \notin \{0, 1\}$ for some index $i$. Therefore, prior work that only considered deterministic routing strategies (Chen et al., 2022; Hu et al., 2024) cannot express the optimal routing strategy and fall back to the near-optimal $s_{\text{MIN}}^{\lambda}$.

**Example** To illustrate routing, consider a scenario with two models $m_1$ and $m_2$. The estimated cost $\hat{c}(x) = (0.9, 1)$ is constant across queries and slightly lower for $m_1$ than for $m_2$. The quality is estimated based on the category of the query, which is either math, code, or generic. For instance, let $\hat{q}(x_{\text{math}}) = (0.8, 0.5)$, $\hat{q}(x_{\text{code}}) = (0.5, 0.8)$, and $\hat{q}(x_{\text{generic}}) = (0.8, 0.9)$. For $\lambda = 1$ and $\gamma = 0.7$, the cost-quality tradeoff is highest for $m_1$, resp. $m_2$, on math, resp. code, queries, and is equal for both models on generic queries. Thus, the router will select $m_1$, resp. $m_2$, for math, resp. code, queries, and select $m_1$ with a probability of $0.7$ and $m_2$ with a probability of $0.3$ for generic queries.

**Ex-Ante Quality Estimation** We explicitly distinguish between the model's true quality $q_i(x)$ and the ex-ante quality estimate $\hat{q}_i(x)$. An optimal routing strategy will select a good model only if $q_i(x) \approx \hat{q}_i(x)$, otherwise the objective in Eq. (1) is not appropriate. Thus, even when routing is suited for the application, the strategy will fail if the quality estimates are inaccurate. This makes quality estimation a critical component of routing strategies. While cost estimation also faces similar challenges, we found that it is less critical and can be approximated more easily.

# 3. Cascading as Sequential Routing

We extend our analysis of the optimal routing strategy to a cascade, providing proofs for all statements in App. B.

**Brief Overview**  In this section, we reinterpret cascading as a sequence of routing problems. Furthermore, we show how our approach improves upon the cascading strategies used in prior work. Finally, we examine the impact of post-hoc quality estimates on the effectiveness of cascading strategies.

**Cascading**  In cascading, an input query is processed sequentially through a chain of models, typically arranged in order of increasing size or cost. The cascade stops once a model's output meets a certain condition, and that output is returned. We will reinterpret cascading as a sequence of routing problems. To do so, we first define the models over which we need to route, which we refer to as *supermodels*.

**Definition 3** (Supermodel). *A supermodel $M$ is a sequence of models $(m_{i_1}, \ldots, m_{i_j})$ such that running a query through $M$ is equivalent to running it through each of the models in the sequence. $\mathcal{M}$ denotes the set of all supermodels and by $M_{i:j}$ we denote the supermodel $(m_i, \ldots, m_j)$.*

In cascading, we only need to consider the supermodels $M_{1:1}, \ldots, M_{1:k}$. The full expressivity of Definition 3 will only be necessary for cascade routing in §4.

**Cascading as Sequential Routing**  Running a cascade on a sample $x$ occurs in a sequence of steps, where at each step, the cascade determines whether to run the next model in the sequence or terminate. By step $j$, we have obtained outputs from the first $j - 1$ models. To decide whether to continue and run $m_j$, we need to determine, in expectation, how well the supermodels $M_{1:j-1}, \ldots M_{1:k}$ will perform on the sample $x$. Once again, this performance is measured as having the highest expected output quality within a certain cost budget. If $M_{1:j-1}$ offers the best performance, we terminate the cascade and return its output, i.e., the output of $m_{j-1}$. Otherwise, if any of $M_{1:j}, \ldots, M_{1:k}$ has better performance, we continue the cascade and run $m_j$. Therefore, at step $j$, the cascade is equivalent to a routing strategy that selects the best supermodel from $M_{1:j-1}, \ldots, M_{1:k}$. Thus, a cascade can be formally defined as follows:

**Definition 4** (Cascading Strategy). *A cascading strategy $s$ is a sequence of routing strategies $(s^{(1)}, \ldots, s^{(k)})$ such that $s^{(j)}$ routes between the supermodels $M_{1:j-1}, \ldots, M_{1:k}$.*

Notably, while the action associated with supermodels $M_{1:j}, \ldots, M_{1:k}$ is the same, namely continuing the cascade, it is important to consider all these supermodels. Indeed, a model $m_j$ might perform poorly while $m_{j+1}$ performs exceptionally well on a given query. In such cases,

the quality-cost tradeoff of $M_{1:j}$ will be worse than the tradeoff of $M_{1:j-1}$, but $M_{1:j+1}$ could still provide a better outcome. Therefore, it is crucial to consider all supermodels $M_{1:j}, \ldots, M_{1:k}$ at step $j$ rather than making decisions solely based on immediate performance.

**Quality and Cost**  To apply Theorem 1 to find the optimal cascading strategy, we first need to derive the quality and cost estimates of the supermodels. Both of these can depend on the answers of previously computed models. Therefore, let $\hat{q}^{(j)}(x)$ and $\hat{c}^{(j)}(x)$ represent the updated estimates in step $j$ after computing the first $j - 1$ models.

We derive the quality and cost estimates associated with supermodel $M_{1:i}$, denoted as $\hat{q}^{(j)}_{1:i}(x)$ and $\hat{c}^{(j)}_{1:i}(x)$, based on the quality and cost estimates of the individual models. Trivially, the cost of the supermodel is equal to the sum of the individual model costs. The quality of a supermodel, however, is governed by the best model within it. Thus, it equals $\mathbb{E}_{\hat{q}_i}[\max(\hat{q}_1(x), \ldots, \hat{q}_i(x))]$, where the expected value reflects the uncertainty in each quality estimate. Specifically, each quality estimate $\hat{q}_i(x)$ is modeled as a random variable estimating the true quality $q_i(x)$. This is crucial since ignoring uncertainty would falsely assume that the quality of a supermodel is always equal to the best model within it, even though the best model may return a poor answer, while another returns a good one. To estimate the uncertainties associated with the estimates, we compute the variance of $\hat{q}^{(j)}_i(x) - \hat{q}^{(k)}_i(x)$ over a validation dataset.

**Optimal Cascading**  We now leverage the optimal routing strategy from Theorem 1 to determine the optimal cascading strategy. As before, optimality is defined in terms of maximizing the expected output quality while adhering to a given cost budget. However, the budget is now only enforced over the entire cascade, and not over individual steps. This leads to a slightly different formulation:

**Theorem 2** (Optimal Cascading Strategy). *For a given cost budget $B$, there exist $\lambda_1, \ldots, \lambda_k \in \mathbb{R}^+$ and a $\gamma \in [0, 1]$ such that the optimal cascading strategy $s_{\mathrm{OPT}} = (s^{(1)}_{\mathrm{OPT}}, \ldots, s^{(k)}_{\mathrm{OPT}})$ is given by $s^{(j)}_{\mathrm{OPT}} = \gamma s^{(j),\lambda_j}_{\mathrm{MIN}} + (1-\gamma)s^{(j),\lambda_j}_{\mathrm{MAX}}$ where $s^{(j),\lambda_j}_{\mathrm{MIN}}$ and $s^{(j),\lambda_j}_{\mathrm{MAX}}$ are defined as in Theorem 1.*

In App. B, we explain how to obtain the hyperparameters $\lambda_1, \ldots, \lambda_k$ and $\gamma$ for a given cost budget $B$ using a validation dataset $D$. In Algorithm 2, we provide pseudocode for the optimal cascading algorithm, illustrating the steps involved in selecting the best model for a given query.

**Example**  To illustrate the optimal cascading strategy, consider once again a scenario with two models $m_1, m_2$ with costs $\hat{c}^{(1)}(x) = (0.5, 1)$ and $\hat{q}^{(1)}(x) = (0.5, 0.8)$ (ignoring uncertainty). In a cascade, we will always run $m_1$ for a query $x$. Thus, we obtain the first model's output.

**Algorithm 2** Optimal Cascading Algorithm

**Input**: input query $x$, current step $j$, quality estimator $\hat{q}_i^{(j)}$, cost estimator $\hat{c}_i^{(j)}$, tradeoff parameters $\lambda_j$ and $\gamma$
**Output**: Whether to stop the cascade or continue with the next model $m_j$

1: **for** $i = j - 1$ **to** $k$ **do**
2:    $\hat{q}_{1:i}^{(j)}(x) := \mathbb{E}_{\hat{q}^{(j)}}[\max(\hat{q}_1^{(j)}(x), \ldots, \hat{q}_i^{(j)}(x))]$
3:    $\hat{c}_{1:i}^{(j)}(x) := \sum_{l=1}^{i} \hat{c}_l^{(j)}(x)$
4: **end for**
5: index := Router$\Big(x, (\hat{q}_{1:j-1}^{(j)}(x), \ldots, \hat{q}_{1:k}^{(j)}(x)),$
6:                     $(\hat{c}_{1:j-1}^{(j)}(x), \ldots, \hat{c}_{1:k}^{(j)}(x)), \lambda_j, \gamma\Big)$
7: **if** index $==$ 1 **then**
8:    **return** stop
9: **else**
10:    **return** run $m_j$
11: **end if**

---

Based on this output, we can adjust the quality and cost estimates. Suppose, for instance, that the model output is very long and its confidence in its own answer is only 30%. Then we update, for instance, $\hat{q}^{(2)}(x) = (0.3, 0.6)$ and $\hat{c}^{(2)}(x) = (1, 2)$. For $\lambda_2 = 1$, we would now stop the cascade and return the output of $m_1$ since the cost-quality tradeoff is highest for the supermodel $\{m_1\}$. If, instead, $\lambda_2 = 0.1$, we would run $m_2$ since the cost-quality tradeoff is highest for the supermodel $\{m_1, m_2\}$. In this case, the cascade would return the output of $m_2$.

**Prior Work** Prior work on cascading has often relied on strong assumptions to simplify the strategy, using a treshold-strategy as an approximation of the optimal cascade. Specifically, in step $j$, the cascade continues if $\hat{q}_{j-1}^{(j)}(x) < \tau_j$ for some threshold $\tau_j \in \mathbb{R}$. To the best of our knowledge, all existing works can be seen as a specific instantiation of this thresholding scheme with cost and quality estimators that depend on the specific application used (Chen et al., 2023; Damani et al., 2024; Gupta et al., 2024; Jitkrittum et al., 2023b; Nie et al., 2024). Below, we outline the conditions under which this simplified approach is optimal.

**Corollary 1** (Optimal Threshold Strategy)**.** *Under minor technical assumptions, the thresholding strategy is equivalent to our cascading strategy if and only if the following conditions hold: $\hat{c}_i^{(j)}(x)$ is independent of $x$ for all $i, j \in \{1, \ldots, k\}$, $\hat{q}_i^{(j)}(x)$ is independent of $x$ for all $i \geqslant j$, and $\hat{q}_{1:i}^{(j)}(x)$ is equal to $\hat{q}_i^{(j)}(x)$.*

**Post-Hoc Quality Estimation** Once again, this theoretical framework highlights the importance of quality estimation, with a shift in focus from ex-ante quality estimation to post-hoc quality estimation, which now plays a critical

role. Cascading approaches are only advantageous when the post-hoc quality estimate provides significantly better information than the ex-ante estimate. If this improvement is minimal, it would be more effective to directly route queries to the most suitable model, bypassing the cascading process. While the post-hoc estimate is essential for refining decisions, the ex-ante quality estimate remains valuable in determining whether future models can potentially deliver better performance. Only in the threshold cascade strategy, where the ex-ante estimate is fixed, does it become irrelevant. By contrast, our approach improves upon the threshold cascade by incorporating both ex-ante and post-hoc quality estimates, thereby enabling more informed decision-making.

## 4.Cascade Routing as Cascade Generalization

Both routing and cascading are powerful techniques that enable the efficient use of multiple models. However, their use is often orthogonal: while routing is useful when ex-ante quality estimates are accurate, cascading is more beneficial when post-hoc estimates are accurate. We therefore present cascade routing, which is a generalization of both techniques. Proofs for all theorems and lemmas in this section are included in App. C.

**Brief Overview** In this section, we first define cascade routing and explain how it generalizes both routing and cascading. We then derive the optimal cascade routing strategy and solve several of the additional challenges that arise when applying cascade routing in practice. Finally, we provide an illustrative example.

**Cascade Routing** Cascade routing closely resembles cascading, but with one crucial difference: the routing strategy at step $j$ routes between all possible supermodels, not just the supermodels $M_{1:j-1}, \ldots, M_{1:k}$. Therefore, both Definition 4 and Theorem 2 can be extended to this setting.

**Definition 5** (Cascade Routing)**.** *A cascade routing strategy $s$ is a sequence of routing strategies $(s^{(1)}, \ldots, s^{(k)})$ such that, for a given sample $x \in \mathcal{X}$, $s^{(j)}$ routes between all supermodels in $\mathcal{M}$ that start with the $j - 1$ models that have already been computed for this query.*

**Theorem 3** (Optimal Cascade Routing)**.** *For a given cost budget $B$, there exist $\lambda_1, \ldots, \lambda_k \in \mathbb{R}^+$ and a $\gamma \in \mathbb{R}^+$ such that the optimal cascade routing strategy $s_{\text{OPT}} = (s_{\text{OPT}}^{(1)}, \ldots, s_{\text{OPT}}^{(k)})$ is given by $s_{\text{OPT}}^{(j)} = \gamma s_{\text{MIN}}^{(j),\lambda_j} + (1-\gamma)s_{\text{MAX}}^{(j),\lambda_j}$ where $s_{\text{MIN}}^{(j),\lambda_j}$ and $s_{\text{MAX}}^{(j),\lambda_j}$ are defined as in Theorem 1.*

In Algorithm 3, we provide pseudocode for the cascade routing algorithm. While cascade routing is a seemingly simple extension of cascading, it also introduces additional challenges which we address now.

**Algorithm 3** Optimal Cascade Routing Algorithm

**Input**: input query $x$, model indices run so far $\{i_1, ..., i_{j-1}\}$, quality estimator $\hat{q}_i^{(j)}$, cost estimator $\hat{c}_i^{(j)}$, tradeoff parameters $\lambda_j$ and $\gamma$

**Output**: Index of the next model to run

1: $\mathcal{S} = [M \subset \{1, \ldots k\} | \forall l \in \{1, \ldots, j-1\} : i_l \in M]$
2: **for** $M \in \mathcal{S}$ **do**
3:     $\hat{q}_M^{(j)}(x) := \mathbb{E}_{\hat{q}^{(j)}}[\max_{l \in M}(\hat{q}_l^{(j)}(x))]$
4:     $\hat{c}_M^{(j)}(x) := \sum_{l \in M} \hat{c}_l^{(j)}(x)$
5: **end for**
6: index := $\text{Router}\Big(x, (\hat{q}_M^{(j)}(x) \text{ for } M \in \mathcal{S}),$
7:                 $(\hat{c}_M^{(j)}(x) \text{ for } M \in \mathcal{S}), \lambda_j, \gamma\Big)$
8: possibilities := $\mathcal{S}_{\text{index}} \setminus \{i_1, ..., i_{j-1}\}$
9: **if** possibilities $= \emptyset$ **then**
10:     **return** stop
11: **else**
12:     min_cost_index := $\arg\min_{i \in \text{possibilities}} c_i(x)$
13:     **return** min_cost_index
14: **end if**

**Model Order** In cascading, the model order is predetermined, and the routing strategy only decides whether to proceed with the next model in the sequence. In contrast, cascade routing must dynamically determine the order in which models are computed. Despite this, both the estimated quality $\hat{q}_M^{(j)}(x)$ and cost $\hat{c}_M^{(j)}(x)$ of a supermodel $M$ are order-independent. Therefore, supermodels that contain the same models in a different order will have the same cost and quality. To mitigate this, we sort the models within the selected supermodel by cost and compute the cheapest one first (illustrated in Lines 12-13 of Algorithm 3). This approach aligns with cascading, where more expensive models are only used if cheaper models do not suffice.

**Number of Supermodels** In cascading, the quality and cost must be computed for a maximum of $k$ supermodels at each step. However, in cascade routing, the number of supermodels grows exponentially, leading to the need to evaluate up to $2^k$ supermodels. This increase can become prohibitively costly, particularly since the model selection process must remain computationally negligible with respect to model computation. To mitigate this, we leverage so-called negative marginal gains. It can be shown (see App. C) that if a model $m$ in a supermodel $M$ negatively impacts the quality-cost tradeoff, all supermodels containing all models in $M$ can be pruned from the search space. For example, if $m_1$ negatively affects the quality-cost tradeoff of the supermodel $\{m_1, m_2\}$, we can prune all supermodels that contain *both* $m_1$ and $m_2$. Since this negative contribution is quite common, this allows us to prune the search space significantly. More formally, this pruning operation relies on the following lemma:

**Lemma 1** (Negative Marginal Gain). *Let $M \in \mathcal{M}$ and $m$ be any model in $M$. Let the marginal gain of $m$ w.r.t. $M$ be defined as $\tau_M(x, \lambda) - \tau_{M \setminus \{m\}}(x, \lambda)$. Then, if the marginal gain of $m$ w.r.t. $M$ is strictly negative for a given query, the optimal cascade routing strategy will never run a supermodel $M' \in \mathcal{M}$ that contains all models in $M$.*

**Example** To illustrate the optimal cascade routing strategy, consider a scenario with two models $m_1$ and $m_2$ with costs $\hat{c}^{(1)}(x) = (0.5, 1)$ and $\hat{q}^{(1)}(x) = (0.5, 0.8)$. In contrast to cascading, we do not necessarily need to run $m_1$ first. In this scenario, if $\lambda_1 = 0.1$, we would immediately run $m_2$. While we would most likely stop after $m_2$, it is possible that its output is so bad that we update the quality estimator to $\hat{q}^{(2)}(x) = (0.5, 0.1)$. In this case, for $\lambda_2 = 0.1$, we would run $m_1$ next and return its output. If $\lambda_1 = 1$, we come into the more classical cascading scenario explained before where $m_1$ is run first.

# 5. Experimental Evaluation

We now evaluate the performance of cascade routing and demonstrate that it significantly outperforms all other strategies. Additionally, we show that our new cascading approach outperforms the threshold-based cascading method. For this purpose, we first conduct experiments on RouterBench (Hu et al., 2024), a benchmark specifically designed to evaluate routing and cascading (§5.1). Next, we test cascade routing on several additional benchmarks to evaluate its performance in more realistic scenarios (§5.2). In the appendix, we perform an ablation study to examine the impact of various design choices in cascade routing on performance and runtime (App. F). Finally, in App. H, we show detailed results as well as cost-quality tradeoff curves for several benchmarks.

### 5.1. RouterBench

RouterBench (Hu et al., 2024) is a benchmark developed to evaluate the efficacy of different model selection strategies. It includes questions from seven diverse benchmarks, such as MMLU (Hendrycks et al., 2021), GSM8k (Cobbe et al., 2021), and MBPP (Austin et al., 2021), alongside answers from eleven different models ranging from GPT-4 (OpenAI, 2023) to Mistral-7B (Jiang et al., 2023).

**Quality and Cost Estimates** Similar to (Hu et al., 2024), we estimate quality and cost by adding zero-centered Gaussian noise to their true values. Both cost and quality estimates are modeled as linear functions fitted on these noisy signals. Thus, the quality estimate can be expressed as $\hat{q}_{W,b}(x) = W(q(x) + \epsilon) + b$ where $\epsilon \sim \mathcal{N}(0, \sigma^2)$. A similar expression holds for the cost estimate. We define the variance of the noisy signal as $\sigma_{\text{ante}}^2$ before model computation (ex-ante estimates) and $\sigma_{\text{post}}^2$ after (post-hoc estimates).

Table 1: AUC scores in % for different strategies on RouterBench across model and noise levels. All baselines are always worse than the 95% confidence intervals of cascade routing. For a discussion on confidence intervals, we refer to App. E.

| | Three Models | | | Five Models | | | Eleven Models | | |
|---|---|---|---|---|---|---|---|---|---|
| | Low | Med | High | Low | Med | High | Low | Med | High |
| Linear Interp. | 69.62 | 69.62 | 69.62 | 69.22 | 69.22 | 69.22 | 70.51 | 70.51 | 70.51 |
| Routing | 79.73 | 74.97 | 71.81 | 81.24 | 74.43 | 71.33 | 83.25 | 74.63 | 72.67 |
| Cascade (Baseline) | 80.86 | 74.64 | 72.48 | 82.33 | 73.03 | 69.53 | 84.48 | 73.64 | 69.79 |
| Cascade (Ours) | 81.09 | 76.16 | 72.67 | 83.06 | 75.17 | 70.18 | 84.47 | 75.10 | 70.26 |
| Cascade Routing (Ours) | **82.36** | **76.55** | **73.22** | **84.33** | **76.31** | **72.75** | **87.24** | **77.57** | **74.40** |

(a) Comparison of cascade routing with routing.

(b) Comparison of cascade routing with cascading.

Figure 2: Difference in AUC performance between cascade routing and baseline strategies on RouterBench for various noise values. Red indicates cascade routing is much better, while blue indicates it is only a bit better.

To explore different uncertainty levels, we vary the variances to simulate low-, medium-, and high-noise scenarios, with exact values for the variances given in App. D.1.

**Models** We evaluate cascade routing on RouterBench using three, five, and eleven models available for model selection, ensuring a comprehensive evaluation across a range of scenarios. The exact models are provided in App. D.1.

**Strategies** We compare cascade routing against several baseline strategies, including the routing strategy described in §2, the threshold-based cascading approach from prior work (Corollary 1), and the optimal cascading strategy (Theorem 2). Additionally, as in (Hu et al., 2024), we include a baseline that linearly interpolates cost and quality on the Pareto frontier of the models.

**Evaluation Metric** For each method, we evaluate performance using cost budgets ranging from the cheapest to the most expensive model. This produces a quality-cost curve for each strategy. Following (Hu et al., 2024), we use the Area Under the Curve (AUC) as the performance metric.

**Results** Table 1 presents the results for the zero-shot setting, with the five-shot results detailed in App. H. Cascade routing consistently outperforms all baseline strategies with performance gains between 1% to 4%, which measured relatively to the naive linear interpolation baseline means that cascade routing improves by 13% to 80% over the baselines. This performance gap widens as more models are available and narrows under higher noise levels, indicating that cascade routing is most effective with large model sets and accurate cost and quality estimates. Furthermore, our new cascading strategy outperforms the threshold-based cascade by up to 2%, reinforcing the practical relevance of our theoretical results.

**Quality Estimation** To better understand the impact of quality estimation on model selection strategies, we additionally conduct experiments with five models under a broader range of varying noise levels. Fig. 2 illustrates the difference in AUC performance between cascade routing and baseline strategies for all possible noise levels. The results demonstrate that cascade routing consistently outperforms the baselines, achieving up to an 8% improvement for cascading and up to a 12% improvement for routing. Notably, the performance gap highlights key differences

between the cascading and routing strategies. For routing, the value of $\sigma_{\text{ante}}$ is critical—high $\sigma_{\text{ante}}$ significantly reduces performance compared to cascade routing. Conversely, for cascading, $\sigma_{\text{post}}$ plays a more influential role, with higher values causing substantial performance degradation. These findings underscore the importance of accurate quality estimation for both strategies. Cascade routing proves to be a more robust solution by unifying the strengths of both approaches and effectively leveraging low $\sigma_{\text{ante}}$ and low $\sigma_{\text{post}}$ to enhance performance.

### 5.2. Real-World Benchmarks

We now show that cascade routing outperforms baselines on more realistic benchmarks with quality estimates that can be used in real-world scenarios. We differentiate in our analysis between benchmarks where accurate quality estimation is available and those where it is not.

**Accurate Quality Estimation**   We first evaluate cascade routing on two benchmarks that allow for accurate quality estimation. First, in the domain of software engineering, it is often easier to generate tests to reproduce specific issues than to fix them. We therefore use SWE-Bench (Jimenez et al., 2024) as a benchmark where accurate post-hoc quality estimation is available. Specifically, we assume that the quality of a model's response can be accurately estimated by testing it on the ground-truth test cases. Second, to simulate a use-case where ex-ante quality estimation is accurate, we use the Math and Coder models from the QWEN-2.5 model family (Yang et al., 2024; Hui et al., 2024) and evaluate them on a combination of Minerva Math (Lewkowycz et al., 2022) and LiveCodeBench (Jain et al., 2024). To obtain accurate quality estimates, we incorporate a sample's origin benchmark as a feature in the quality estimation model. For all details about the benchmarks, models, and estimators, we refer to App. D.1.

**Results**   Table 2 (left) shows the results for both benchmarks. In SWE-Bench, our methods outperform baseline strategies by up to $14\%$. As expected, the routing strategy does not outperform the trivial baseline on this benchmark, as ex-ante quality estimates are insufficient. Interestingly, despite perfect post-hoc quality estimation for SWE-Bench, the baseline cascade strategy also performs poorly. This is due to the binary feedback of the quality estimator, which leads the threshold $\tau$ of the baseline cascade to either admit all models ($\tau = 0$) or only correct ones ($\tau > 0$).

For Minerva Math and LiveCodeBench, the opposite trend holds true. With accurate ex-ante quality estimation, the routing strategy achieves strong performance, surpassing the baseline cascade strategy by $10\%$. However, the cascade routing strategy still outperforms all methods, highlighting its robustness across diverse benchmarks and

quality estimation scenarios. Interestingly, despite poor post-hoc quality estimation, our cascading strategy nearly matches the performance of routing. This suggests that the cascade effectively leverages ex-ante quality estimation to make informed decisions, unlike the baseline cascade.

We highlight that the cost estimator for SWE-Bench is latency-based, computing cost as the time it takes to complete the task. In contrast, the estimator for Minerva Math and LiveCodeBench uses the cost of the generation. Thus, cascade routing can adapt to different cost estimators.

**Poor Quality Estimation**   We perform experiments on classification and open-form reasoning tasks where there is no known accurate quality estimator. The classification benchmarks include ARC-Challenge (Clark et al., 2018), MMLU-Pro (Wang et al., 2024), and MixEval (Ni et al., 2024). For open-form reasoning tasks, we use MMLU-Pro and GSM8k (Cobbe et al., 2021). In classification, models select a single option representing their answer, with no intermediate reasoning process. In contrast, open-form reasoning allows models to generate their answers after reasoning. In this section, we evaluate two model families consisting of three models, LLAMA and GEMMA, and show similar numbers for the MISTRAL model family in App. H. We create a quality estimator based on state-of-the-art work Gupta et al. (2024), which uses log probabilities as features. For full details on the benchmarks, models, and cost and quality estimators, we refer to App. D.

**Results**   Table 2 (right) presents the results for the LLAMA and GEMMA model families across both benchmarks. Cascade routing consistently performs on par with or outperforms all baselines, though with much narrower margins reaching up to $1.2\%$. This reduced gain can be attributed to the fact that the quality and cost estimates are very noisy, leading to performance gains over the naive baseline similar to those observed in very high-noise scenarios on RouterBench.

## 6. Related Work

**Routing**   Routing is a widely studied problem in machine learning, particularly in the task of directing input queries to specialized models. One of the most common applications of routing is model selection for natural language input queries with a known answer (Chuang et al., 2024; Ding et al., 2024; Hari & Thomson, 2023; Liu et al., 2024; Jang et al., 2023; Nguyen et al., 2024; Sakota et al., 2024; Shnitzer et al., 2023). All these works train a model to predict whether a given model will correctly answer a query. Though the setups in these works are largely similar, they vary in certain specifics, such as the type of input queries or the features used for quality estimation.

Table 2: AUC scores on practical benchmarks. On the left, resp. right, side we show the benchmarks with good, resp. poor, quality estimates. The highest numbers are bolded, and underlined numbers are within the $95\%$ confidence intervals of the highest number. For a discussion on confidence intervals, refer to App. E. In App. G, we present benchmark-specific AUC values for results averaged over several benchmarks.

| | SWE-Bench | | Math+Code | Classification | | Open-Form | |
|---|---|---|---|---|---|---|---|
| | 10 MODELS | 5 MODELS | QWEN | LLAMA | GEMMA | LLAMA | GEMMA |
| Linear Interp. | 40.51 | 38.64 | 39.63 | 74.28 | 61.68 | 79.11 | 54.10 |
| Routing | 40.47 | 39.40 | 47.46 | 74.92 | 64.44 | 79.32 | 58.40 |
| Cascade (Baseline) | 38.52 | 45.89 | 37.68 | 74.81 | 54.32 | 79.23 | 56.18 |
| Cascade (Ours) | 53.20 | 50.94 | 46.76 | 75.46 | 62.79 | 79.22 | 56.18 |
| Cascade Routing (Ours) | **54.12** | **51.09** | **48.55** | **75.52** | **64.84** | **79.88** | **59.66** |

Routing is also applied in other areas. For instance, Lu et al. (2024); Ong et al. (2024) use preference data to train a quality estimator, which facilitates routing in scenarios involving real-world user queries where clear ground-truth answers may not exist. Additionally, Chen et al. (2022) employ routing for API selection in multi-label classification tasks, focusing on directing queries to the appropriate API based on task requirements. Similarly, Zhang et al. (2024b) apply routing in software agent environments, directing user issues to the agent most suited to handle them. Finally, Pichlmeier et al. (2024) dynamically routes token generation instead of entire queries, allowing for more fine-grained routing decisions.

**Cascading** Cascading techniques are primarily used to reduce inference costs by employing smaller models initially and only cascading to larger models if the smaller ones fail to provide a sufficiently accurate answer. Most often, cascading decisions are based on the smaller model's confidence in its own predictions (Chen et al., 2023; 2024; Ramírez et al., 2024; Varshney & Baral, 2022). However, alternative techniques also exist. For example, Madaan et al. (2023) propose running models multiple times and measuring the variance in their responses to decide whether to cascade to a larger model.

For classification tasks, early stopping is another cascading strategy (Li et al., 2021; Schuster et al., 2022). In this approach, the cascade halts when a model's intermediate layers generate representations that are sufficiently informative to predict the correct class. This reduces computational costs by avoiding the need to process every query through the entire model.

There has also been specific research on quality estimation within cascading frameworks. Gupta et al. (2024) examine various measures of uncertainty in language model answers, evaluating their impact on cascading performance. Meanwhile, Jitkrittum et al. (2023a) explore failure cases in cascading mechanisms that rely on uncertainty, introducing alternative quality measures that enhance cascade effi-

ciency. Furthermore, Xue et al. (2023) apply cascading to majority voting for a single model to obtain a method called dynamic voting: the cascade stops depending on the aggregated answers of all previous model computations. Lastly, (Zhang et al., 2024a) propose the use of multi-objective quality metrics to guide cascading decisions and do not solely focus on accuracy.

All works mentioned here can be seen as an instantiation of the thresholding mechanism outlined in Corollary 1 with application-specific quality and cost estimates.

## 7. Conclusion

In this work, we introduced a novel framework for routing and cascading that enabled us to propose theoretically optimal strategies for both paradigms. Further, we used this analysis to propose a new paradigm for model selection, cascade routing, which combines the benefits of routing and cascading. We showed that cascade routing can significantly outperform its baselines, especially with good quality and cost estimates. We also find that our new cascading strategy significantly outperforms existing approaches to cascading, showing our theoretical analysis also leads to practical gains.

## Impact Statement

Our work can significantly impact the field of model selection strategies. By providing a theoretical foundation for routing and cascading, we have shown that these strategies can be improved by using more accurate quality and cost estimates. Cascade routing combines the strengths of both routing and cascading and offers a more flexible and effective model selection strategy. Furthermore, by underscoring the importance of quality estimation, we highlighted a critical area for future research in model selection strategies that could lead to further improvements in this area.

## Acknowledgements

This work was funded in part by the Swiss National Science Foundation (SNSF) [200021_207967].

This work has been done as part of the EU grant ELSA (European Lighthouse on Secure and Safe AI, grant agreement no. 101070617). Views and opinions expressed are however those of the authors only and do not necessarily reflect those of the European Union or European Commission. Neither the European Union nor the European Commission can be held responsible for them.

The work has received funding from the Swiss State Secretariat for Education, Research and Innovation (SERI).

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

# A. Routing

First, we explain how to obtain the hyperparameters $\lambda$ and $\gamma$ for the routing strategy. We then provide a more exact formulation of the routing optimization problem and prove Theorem 1.

**Hyperparameters** Due to the second part of Theorem 1, we only need to find *a* set of hyperparameters $\lambda$ and $\gamma$ that achieve the cost budget. Indeed, all routing strategies that have an expected cost that is exactly equal to $B$ and can be written as a convex combination of $s_{\text{MIN}}^{\lambda'}$ and $s_{\text{MAX}}^{\lambda'}$ for some $\lambda' \in \mathbb{R}^+$ achieve the same optimal quality.

To determine these parameters, we estimate the cost of a strategy using a validation dataset $D$ that is representative of the query distribution $\mathcal{X}$. We then perform a hyperparameter search to find optimal values of $\lambda$ and $\gamma$. By leveraging several properties of routing strategies, one can show that this hyperparameter search can be reduced to a single binary search over $\lambda$, enabling a quick and efficient hyperparameter optimization process.

**Proving the Theorem** To prove Theorem 1, we first rewrite the routing optimization problem in Eq. (1) as a linear program over functions $s : \mathcal{X} \to \mathbb{R}^k$ instead of functions $s : \mathcal{X} \to \Delta_k$. This makes the optimization problem more tractable. Specifically, Eq. (1) can be rewritten as follows:

$$
\begin{aligned}
\max_{r} \quad & \mathbb{E}_{x \sim \mathcal{X}} \left[ \sum_{i=1}^{k} s_i(x) \hat{q}_i(x) \right] \\
\text{s.t.} \quad & \mathbb{E}_{x \sim \mathcal{X}} \left[ \sum_{i=1}^{k} s_i(x) \hat{c}_i(x) \right] \leqslant B \\
& \forall i \in \{1, ..., k\} : \forall x \in \mathcal{X} : s_i(x) \geq 0 \wedge \sum_{j=1}^{k} s_j(x) = 1
\end{aligned}
\tag{2}
$$

We then rewrite Theorem 1 to allow for a more exact formulation of the optimal routing strategy:

**Theorem 4.** *(Optimal Routing Strategy) Suppose there exists an admissible solution to the set of constraints in Eq. (2). For any $\lambda \in \mathbb{R}^+$, let $S_\lambda$ be the set of routing strategies $s$ that satisfy the following constraints:*

$$
\forall x \in \mathcal{X}, \forall i \in \{1, ..., k\} : \hat{q}_i(x) - \lambda \hat{c}_i(x) < \max_j \hat{q}_j(x) - \lambda \hat{c}_j(x) \Rightarrow s_i(x) = 0
\tag{3}
$$

*If there exists a strategy in $S_0$ that has a cost less than or equal to $B$, then this strategy achieves the optimal quality. Otherwise, there exists a $\lambda^* \in \mathbb{R}^+$ such that $S_\lambda$ contains a routing strategy that has exactly cost $B$ and all routing strategies in $\bigcup_{\lambda \in \mathbb{R}^+} S_\lambda$ that have cost $B$ achieve the same optimal quality.*

There is one extra condition mentioned here that we omitted in the main text. The requirement of having at least an admissible solution to the constraints in Eq. (2) is necessary to ensure that the set of possible solutions to Eq. (2) is not empty. For instance, the cost budget $B$ can be too low such that even running the cheapest model for each query is too expensive.

The formulation of $s_{\text{OPT}}$ as a convex combination of $s_{\text{MIN}}^\lambda$ and $s_{\text{MAX}}^\lambda$ is a direct consequence of Theorem 4. Indeed, let $\lambda^*$ be as defined in Theorem 4. Then $s_{\text{MIN}}^{\lambda^*}$, resp. $s_{\text{MAX}}^{\lambda^*}$, must have the lowest, resp. highest, cost among all routing strategies in $S_{\lambda^*}$. Since there is a routing strategy in $S_{\lambda^*}$ that has cost $B$, there must exist a convex combination of $s_{\text{MIN}}^{\lambda^*}$ and $s_{\text{MAX}}^{\lambda^*}$ that also has cost $B$ and thus achieves the optimal quality.

We first prove several lemmas before proving the theorem.

**Lemma 2.** *$S_\lambda$ is non-empty and convex for all $\lambda \in \mathbb{R}^+$.*

*Proof.* Non-emptiness follows from the fact that the routing strategy that assigns all probability mass for a sample $x$ to a model $i$ for which $\hat{q}_i(x) - \lambda \hat{c}_i(x)$ is maximal, is in $S_\lambda$. For convexity, let $s^{(1)}, s^{(2)} \in S_\lambda$ be arbitrary. Let $s^\gamma$ be the convex combination of $s^{(1)}$ and $s^{(2)}$ with weight $\gamma \in [0, 1]$. Let $x \in \mathcal{X}$ be arbitrary. Then, $s_i^\gamma(x) > 0$ if and only if $s_i^{(1)}(x) > 0$ or $s_i^{(2)}(x) > 0$. Since $s^{(1)}, s^{(2)} \in S_\lambda$, we have $\hat{q}_i(x) - \lambda \hat{c}_i(x) \geqslant \max_j \hat{q}_j(x) - \lambda \hat{c}_j(x)$ for all $i$ such that $s_i^{(1)}(x) > 0$ or $s_i^{(2)}(x) > 0$. This implies that $\hat{q}_i(x) - \lambda \hat{c}_i(x) \geqslant \max_j \hat{q}_j(x) - \lambda \hat{c}_j(x)$ for all $i$ such that $s_i^\gamma(x) > 0$. Thus, $s^\gamma \in S_\lambda$. $\square$

**Lemma 3.** *Let $\lambda_1 < \lambda_2$ and $s^{(1)}$, resp. $s^{(2)}$ be arbitrary routing strategies in $S_{\lambda_1}$, resp. $S_{\lambda_2}$. Then, the cost of $s^{(1)}$ is greater or equal to the cost of $s^{(2)}$, i.e.,*

$$\mathbb{E}_{x \sim \mathcal{X}}\left[\sum_{i=1}^{k} s_i^{(1)}(x)\hat{c}_i(x)\right] \geqslant \mathbb{E}_{x \sim \mathcal{X}}\left[\sum_{i=1}^{k} s_i^{(2)}(x)\hat{c}_i(x)\right]$$

*Proof.* We show that for any $x \in \mathcal{X}$, the cost of $s^{(1)}$ is greater or equal to the cost of $s^{(2)}$. Let $x \in \mathcal{X}$ be arbitrary. Suppose $s^{(1)}$ is strictly cheaper than $s^{(2)}$. Then, there must exist a model pair $i, j$ such that $\hat{c}_i(x) < \hat{c}_j(x)$, $s_i^{(1)}(x) > s_i^{(2)}(x) \geqslant 0$, and $s_j^{(2)}(x) > s_j^{(1)}(x) \geqslant 0$. However, $s_i^{(1)}(x) > 0$ implies

$$\hat{q}_i(x) - \lambda_1 \hat{c}_i(x) \geqslant \hat{q}_j(x) - \lambda_1 \hat{c}_j(x).$$

Furthermore, since $\lambda_1 - \lambda_2 < 0$, we have

$$\hat{c}_i(x)(\lambda_1 - \lambda_2) > \hat{c}_j(x)(\lambda_1 - \lambda_2).$$

Adding these two inequalities gives

$$\hat{q}_i(x) - \lambda_2 \hat{c}_i(x) > \hat{q}_j(x) - \lambda_2 \hat{c}_j(x),$$

which is a contradiction with $s_j^{(2)}(x) > 0$. Thus, the cost of $s^{(1)}$ is greater or equal to the cost of $s^{(2)}$. $\qquad\square$

**Lemma 4.** *Let $\Lambda$ be the set of points $\lambda \in \mathbb{R}$ such that there exist an $x \in \mathcal{X}$ and $i \neq j$ such that $\hat{q}_i(x) - \lambda \hat{c}_i(x) = \hat{q}_j(x) - \lambda \hat{c}_j(x)$. Let $\lambda_1 < \lambda_2$ be such that $[\lambda_1, \lambda_2] \cap \Lambda = \emptyset$. Then, $S_{\lambda_1} = S_{\lambda_2}$. Furthermore, if $[\lambda_1, \lambda_2] \cap \Lambda = \{\lambda^*\}$, then $S_\lambda \subset S_{\lambda^*}$ for all $\lambda \in [\lambda_1, \lambda_2]$.*

*Proof.* We first show the first statement by showing that $S_{\lambda_1} \setminus S_{\lambda_2} = \emptyset$. $S_{\lambda_2} \setminus S_{\lambda_1} = \emptyset$ follows analogously. Suppose there exists a routing strategy $s \in S_{\lambda_1} \setminus S_{\lambda_2}$. Since $s \notin S_{\lambda_2}$, there must exist an $x \in \mathcal{X}$ and model $i$ such that $s_i(x) > 0$ and $\hat{q}_i(x) - \lambda_2 \hat{c}_i(x) < \max_j \hat{q}_j(x) - \lambda_2 \hat{c}_j(x)$. Let $j$ be an index such that $\hat{q}_i(x) - \lambda_2 \hat{c}_i(x) < \hat{q}_j(x) - \lambda_2 \hat{c}_j(x)$. Since $s \in S_{\lambda_1}$, we have $\hat{q}_i(x) - \lambda_1 \hat{c}_i(x) \geqslant \hat{q}_j(x) - \lambda_1 \hat{c}_j(x)$. By continuity, there exists a $\lambda \in [\lambda_1, \lambda_2]$ such that $\hat{q}_i(x) - \lambda \hat{c}_i(x) = \hat{q}_j(x) - \lambda \hat{c}_j(x)$, which is a contradiction with $[\lambda_1, \lambda_2] \cap \Lambda = \emptyset$.

Now suppose $[\lambda_1, \lambda_2] \cap \Lambda = \{\lambda^*\}$. Let $\lambda \in [\lambda_1, \lambda^*)$ be arbitrary and let $s \in S_\lambda$ be arbitrary. We show that $s \in S_{\lambda^*}$. For $\lambda \in (\lambda^*, \lambda_2]$, the proof is completely analogous. By contradiction, suppose there exists an $x \in \mathcal{X}$ and model $i$ such that $s_i(x) > 0$ and $\hat{q}_i(x) - \lambda^* \hat{c}_i(x) < \max_j \hat{q}_j(x) - \lambda^* \hat{c}_j(x)$. This means there exists a model $j$ such that $\hat{q}_i(x) - \lambda^* \hat{c}_i(x) < \hat{q}_j(x) - \lambda^* \hat{c}_j(x)$. Since $s \in S_\lambda$, we know that $\hat{q}_i(x) - \lambda \hat{c}_i(x) \geqslant \hat{q}_j(x) - \lambda \hat{c}_j(x)$. This implies that there must exist a $\lambda' \in [\lambda_1, \lambda^*)$ such that $\hat{q}_i(x) - \lambda' \hat{c}_i(x) = \hat{q}_j(x) - \lambda' \hat{c}_j(x)$. However, this is a contradiction with $[\lambda_1, \lambda^*) \cap \Lambda = \emptyset$. Thus, $s \in S_{\lambda^*}$. $\qquad\square$

In what follows, we will assume that $|\Lambda| < \infty$. This is a very minor assumption. For instance, if $\hat{q}$ and $\hat{c}$ only take on a finite amount of values, this is trivially satisfied. Since estimators are implemented on a computer, they will always have a finite precision, meaning that $\hat{q}$ and $\hat{c}$ will only take on a finite amount of values.

**Lemma 5.** *Let $\lambda_1 < \lambda_2$ and $s^{(1)}$, resp. $s^{(2)}$ be arbitrary routing strategies in $S_{\lambda_1}$, resp. $S_{\lambda_2}$, with costs resp. $B_1$ and $B_2$. Then, for any $B \in [B_1, B_2]$ there exists a $\lambda \in [\lambda_1, \lambda_2]$ such that $S_\lambda$ contains a routing strategy that has exactly cost $B$.*

*Proof.* Let $B \in [B_1, B_2]$ be arbitrary. If $B = B_1$ or $B = B_2$, the statement is trivially true. Therefore, suppose $B \in (B_1, B_2)$. Let $\Lambda$ be as defined in Lemma 4. By Lemma 3, there exists a $\lambda^* \in [\lambda_1, \lambda_2]$ such that all strategies in $S_\lambda$ for $\lambda < \lambda^*$, resp. $\lambda > \lambda^*$, have cost at least, resp. at most, $B$. If $\lambda^* \notin \Lambda$, then the first part of Lemma 4, together with $|\Lambda| < \infty$, implies that $S_{\lambda^*} = S_{\lambda^* - \epsilon} = S_{\lambda^* + \epsilon}$ for some $\epsilon > 0$. All the strategies in $S_{\lambda^*}$ must therefore have cost both at least and at most $B$, meaning they should equal $B$. We can therefore assume that $\lambda^* \in \Lambda$. By Lemma 4 and $|\Lambda| < \infty$, there is en $\epsilon > 0$ such that $S_{\lambda^* - \epsilon} \subset S_{\lambda^*}$ and $S_{\lambda^* + \epsilon} \subset S_{\lambda^*}$. Let $s^- \in S_{\lambda^* - \epsilon}$ and $s^+ \in S_{\lambda^* + \epsilon}$ be arbitrary. Let $s^\gamma$ be the convex combination of $s^-$ and $s^+$ with weight $\gamma \in [0, 1]$. Since $s^-, s^+ \in S_{\lambda^*}$, we have $s^\gamma \in S_{\lambda^*}$ by Lemma 2. Denote by $B^-$, resp. $B^+$, the cost of $s^-$, resp $s^+$. Furthermore, the cost of $s^\gamma$ is $\gamma B^- + (1 - \gamma)B^+$. Since $B \in [B^-, B^+]$, there exists a $\gamma \in [0, 1]$ such that $s^\gamma$ has cost exactly $B$. $\qquad\square$

We can now prove the theorem.

*Proof.* If $S_0$ contains a solution that has cost less than or equal to $B$, then this solution trivially achieves the optimal quality. Thus, for the rest of the proof we can assume that the cost of every solution in $S_0$ is greater than $B$. For $\lambda \to \infty$, $S_\lambda$ contains the solution that assigns all probability mass to the model with the lowest cost. Since there is an admissible solution, this solution necessarily has cost less than $B$. Therefore, by Lemma 5, there exists a $\lambda^* \in \mathbb{R}$ such that $S_{\lambda^*}$ contains a routing strategy that has exactly cost $B$.

Let $s$ be an arbitrary routing strategy in $\bigcup_{\lambda \in \mathbb{R}^+} S_\lambda$ that has cost $B$. Specifically, let $s \in S_\lambda$. Let $s'$ be any other routing strategy that is an admissible solution to the optimization problem. Then:

$$
\begin{aligned}
\mathbb{E}_{x \in X} \left[ \sum_{i=1}^{k} s_i'(x) \hat{q}_i(x) \right] &= \mathbb{E}_{x \in X} \left[ \sum_{i=1}^{k} s_i'(x) \hat{q}_i(x) - \lambda B + \lambda B \right] \\
&\leqslant \mathbb{E}_{x \in X} \left[ \sum_{i=1}^{k} s_i'(x) \left( \hat{q}_i(x) - \lambda \hat{c}_i(x) \right) + \lambda B \right] \\
&\leqslant \mathbb{E}_{x \in X} \left[ \sum_{i=1}^{k} s_i(x) \left( \hat{q}_i(x) - \lambda \hat{c}_i(x) \right) + \lambda B \right] \\
&= \mathbb{E}_{x \in X} \left[ \sum_{i=1}^{k} s_i(x) \hat{q}_i(x) \right]
\end{aligned}
$$

Thus, $s$ achieves the optimal quality.

$\square$

## B. Cascading

To prove Theorem 2, we heavily rely on the results derived in App. A. As explained in §3, cascading can be reinterpreted as a sequence of routing problems. However, to prove optimality, we need to be slightly more careful with the exact formulation of the problem.

At step $j$, the cascading strategy needs to decide whether to stop the cascade or to continue to the next model. It should continue to the next model if any of the supermodels $M_{1:j}, \ldots, M_{1:k}$ is better to run than $M_{1:j-1}$ for some measure of 'better'. Therefore, the cascading strategy is indeed performing a routing operation between the supermodels $M_{1:j-1}, \ldots, M_{1:k}$.

However, the optimization problem does slightly change compared to the routing problem. First of all, for each query $x \in \mathcal{X}$, there is a possibility that the cascade is stopped before step $j$. Therefore, the cascade should not aim to optimize the quality at step $j$ for such a query, since it would not have any effect on the overall quality of the cascade. Furthermore, the budget $B$ is only enforced over the entire cascade, and not over the individual steps. Since the problem changes through steps, it is not required that the cost of the router at step $j$ is exactly equal to $B$.

Therefore, we reformulate cascading using an inner and outer optimization problem. The inner optimization problem aims to find the optimal routing strategy at step $j$ for a given budget $B_j$. The outer optimization problem aims to find the optimal budget $B_j$ for each step $j$ such that the overall quality of the cascade is maximized under the constraint that the total cost of the cascade is at most $B$.

To formulate this more exactly, let $P_j(M)$ be the probability that the cascade computed supermodel $M$ by step $j$. Then, the inner optimization problem at step $j$ can be formulated as:

$$\max_{r^{(j)}} \quad \mathbb{E}_{x \sim \mathcal{X}} \left[ P_j(M_{1:j-1}) \sum_{i=j-1}^{k} r_{1:i}(x) \hat{q}_{1:i}^{(j)}(x) \right]$$

$$\text{s.t.} \quad \mathbb{E}_{x \sim \mathcal{X}} \left[ P_j(M_{1:j-1}) \sum_{i=j-1}^{k} r_{1:i}(x) \hat{c}_{1:i}^{(j)}(x) \right] \leqslant B_j \tag{4}$$

$$\forall i \in \{j-1, ..., k\} : \forall x \in \mathcal{X} : r_{1:i}(x) \geq 0 \wedge \sum_{i=j-1}^{k} r_{1:i}(x) = 1$$

Note that $P_j(M_{1:j-1})$ can be incorporated in the quality and cost estimates. This leaves us with the exact same optimization problem as the routing problem, but with a different budget $B_j$. Since the chosen model only depends on the maximization of $P_j(M_{1:j-1})\hat{q}_i^{(j)}(x) - \lambda_j P_j(M_{1:j-1})\hat{c}_i^{(j)}(x)$, the probability $P_j(M_{1:j-1})$ can be divided out of the optimization problem.

The inner optimization problems prove the existence of optimal routing strategies at each step $j$ with parameters $\lambda_j$. We note that there only needs to be one parameter $\gamma$ that determines the convex combination since the budget $B$ is only enforced over the entire cascade.

Let us denote the quality and cost of the entire cascading strategy for given parameters $\lambda_1, \ldots, \lambda_k$ and $\gamma$ as $Q(\lambda_1, \ldots, \lambda_k, \gamma)$ and $C(\lambda_1, \ldots, \lambda_k, \gamma)$ respectively. Then, the outer optimization problem can be formulated as:

$$\max_{\lambda_1,...,\lambda_k,\gamma} \quad Q(\lambda_1, \ldots, \lambda_k, \gamma)$$

$$\text{s.t.} \quad C(\lambda_1, \ldots, \lambda_k, \gamma) \leqslant B \tag{5}$$

To solve this outer optimization problem, we simply perform a hyperparameter search over the budgets $B_1, \ldots, B_k$ using a hyperparameter optimization search as discussed in §3.

## B.1. Prior Approximations

We now prove Corollary 1. Before doing so, we first need to define what we exactly mean by equivalency. For this purpose, let $\mathcal{C}_1$ be defined as follows:

$$\mathcal{C}_1 = \left\{ s \mid s \text{ is a cascading strategy with parameters } \lambda_1, \ldots, \lambda_k, \gamma = 0 \text{ using estimates } \hat{q}^{(j)}, \hat{c}^{(j)} \right\}$$

Similarly, let $\mathcal{C}_2$ be defined as follows:

$$\mathcal{C}_2 = \left\{ s \mid s \text{ is a thresholding strategy with parameters } \tau_1, \ldots, \tau_k \text{ using estimates } \hat{q}^{(j)}, \hat{c}^{(j)} \right\}$$

We note that we set $\gamma = 0$ since the thresholding strategy is deterministic. We therefore restrict the cascading strategy to be deterministic as well.

We define the equivalence between the two sets as follows:

**Definition 6** (Equivalence of Strategies). *We say a set of strategies $\mathcal{C}_1$ is equivalent to another set of strategies $\mathcal{C}_2$, denoted as $\mathcal{C}_1 \equiv \mathcal{C}_2$, if for all $s_0 \in \mathcal{C}_1 \cup \mathcal{C}_2$ there exists a $s_1 \in \mathcal{C}_1$, and a $s_2 \in \mathcal{C}_2$ such that for all $x \in \mathcal{X}$, $s_0$, $s_1$ and $s_2$ take the same decisions on $x$.*

We can now more accurately state the conditions under which the thresholding strategy is equivalent to the optimal strategy.

**Corollary 2** (Optimal Thresholding Strategy). *Let $\mathcal{C}_1$, $\mathcal{C}_2$ be defined as above. Then, $\mathcal{C}_1 \equiv \mathcal{C}_2$ if and only if there exists alternative quality and cost estimates $\hat{q}_i^{(j)'}(x)$ and $\hat{c}_i^{(j)'}(x)$ with associated set of cascading strategies $\mathcal{C}_1'$ such that $\mathcal{C}_1 \equiv \mathcal{C}_1'$ and the following conditions hold on these alternative quality and cost estimates: $\hat{c}_i^{(j)'}(x)$ is independent of $x$ and bigger than 0, $\hat{q}_i^{(j)'}(x)$ is independent of $x$ for all $i \geqslant j$, and $\hat{q}_{1:i}^{(j)'}(x)$ is equal to $\hat{q}_i^{(j)'}(x)$.*

The main difference between Corollary 2 and Corollary 1 is that we impose the possibility of alternative quality and cost estimates. However, this does not really influence equivalency in the intuitive sense. Indeed, one could alternatively phrase

the corollary as follows: the thresholding strategy is equivalent to any of our cascading strategies if and only if it is possible to construct alternative estimates such that the conditions hold.

*Proof.* We note that the cascade $s \in \mathcal{C}_1$ continues on a sample if the following condition holds:

$$\hat{q}_{1:j-1}^{(j)}(x) - \lambda_j \hat{c}_{1:j-1}^{(j)}(x) < \max_{i \in \{j,\dots,k\}} \hat{q}_{1:i}^{(j)}(x) - \lambda_j \hat{c}_{1:i}^{(j)}(x) \tag{6}$$

If $\mathcal{C}_1 \equiv \mathcal{C}_1'$, it is clear that Eq. (6) reduces to the thresholding strategy for all strategies in $\mathcal{C}_1'$. Indeed, for any $s \in \mathcal{C}_1'$, set $\tau_j = \max_{i \in \{j,\dots,k\}} \hat{q}_{1:i}^{(j)} - \lambda_j \hat{c}_{j:i}^{(j)}$ and the thresholding strategy is equivalent to $s$. Alternatively, if $s \in \mathcal{C}_2$, suppose $\max_{i \in \{j,\dots,k\}} \hat{q}_{1:i}^{(j)} - \lambda_j \hat{c}_{j:i}^{(j)} = \hat{q}_{1:i}^{(j)} - \lambda_j \hat{c}_{j:i}^{(j)}$ for some index $i$. Then, set $\lambda_j = \tau_j / \hat{c}_{j:i}^{(j)} - \hat{q}_{1:i}^{(j)} / \hat{c}_{j:i}^{(j)}$ and the cascading strategy is equivalent to $s$. Therefore, $\mathcal{C}_1 \equiv \mathcal{C}_1' \equiv \mathcal{C}_2$.

Suppose now that $\mathcal{C}_1 \equiv \mathcal{C}_2$. We construct alternative quality and cost estimates $\hat{q}_i^{(j)'}(x)$ and $\hat{c}_i^{(j)'}(x)$ such that the conditions hold and such that $\mathcal{C}_1 \equiv \mathcal{C}_1'$. For this purpose, we define $\hat{c}_i^{(j)'}(x) = 1$ for all $i, j \in \{1, \dots, k\}$, $\hat{q}_i^{(j)'}(x) = 1$ for all $i \geqslant j$, and $\hat{q}_i^{(j)'}(x) = \hat{q}_i^{(j)}(x)$ otherwise. Furthermore, we set $\hat{q}_{1:i}^{(j)'}(x) = \hat{q}_i^{(j)'}(x)$ for all $i, j \in \{1, \dots, k\}$. The equivalence of $\mathcal{C}_1'$ and $\mathcal{C}_2$ can now be proven analogously to the previous paragraph. Therefore, $\mathcal{C}_1 \equiv \mathcal{C}_1' \equiv \mathcal{C}_2$. $\qquad\square$

# C. Cascade Routing

We first note that the proof of the optimality of the cascade routing strategy is equivalent to the proof of the optimality of the cascade strategy, except that the expectation in the optimization problem Eq. (4) is now not only over $x \in X$, but also over all possible supermodels that were computed by step $j - 1$. However, this does not change the optimization problem, and the proof is completely analogous to the proof given in §3. Thus, all we need to prove is Lemma 1. To prove the lemma, we first prove the following lemma.

**Lemma 6.** *Let $Q_1, \dots, Q_k$ be distributions. Let $\mathcal{S}$ be the superset of $\{1, \dots, k\}$. Then $f : \mathcal{S} \to \mathbb{R}$ defined as $f(S) = \mathbb{E}(\max_{i \in S} Q_i)$ is submodular. Here, we define $\max_{i \in \emptyset} Q_i = -\infty$*

*Proof.* Let $T \subset S \subset \{1, \dots, k\}$ and $j \in \{1, \dots, k\}$ be arbitrary. To show the submodularity of $f$, we need to show that

$$f(T \cup \{j\}) - f(T) \geq f(S \cup \{j\}) - f(S).$$

We can write:

$$\begin{aligned}
f(S \cup \{j\}) - f(S) &= \mathbb{E}(\max_{i \in S \cup \{j\}} Q_i) - \mathbb{E}(\max_{i \in S} Q_i) \\
&= \mathbb{E}(\max(0, Q_j - \max_{i \in S} Q_i)) \\
&\leqslant \mathbb{E}(\max(0, Q_j - \max_{i \in T} Q_i)) \\
&= \mathbb{E}(\max_{i \in T \cup \{j\}} Q_i) - \mathbb{E}(\max_{i \in T} Q_i) \\
&= f(T \cup \{j\}) - f(T).
\end{aligned}$$

In the proof, we needed $\max_{i \in \emptyset} Q_i = -\infty$ in the case $T = \emptyset$. $\qquad\square$

We note that the assertion that $\max_{i \in \emptyset} Q_i = -\infty$ corresponds to the fact that giving no answer to a query has $-\infty$ quality.

We can now prove Lemma 1.

*Proof.* Let $M$ and $m$ be as in the lemma. Suppose $M'$ is a supermodel that contains all models in $M$. Furthermore, let $M'' = M' \setminus m$. We show that the supermodel $M''$ is always strictly preferred over $M'$. To see this, we note that the difference between $\tau_{M'}(x, \lambda)$ and $\tau_{M''}(x, \lambda)$ is equal to

$$\mathbb{E}(\max_{m' \in M'} \hat{q}_{m'}(x)) - \mathbb{E}(\max_{m' \in M''} \hat{q}_{m'}(x)) - \lambda_j \hat{c}_m(x)$$

By Lemma 6, this difference is smaller than $\hat{q}_M(x) - \hat{q}_{M \setminus \{m\}}(x) - \lambda_j \hat{c}_m(x)$. Thus, by assumption, this difference is negative, and therefore $M''$ is always preferred over $M'$, which concludes the proof. $\qquad\square$

Table 3: Standard deviations of the noise levels on the RouterBench dataset.

| | Quality | | Cost | |
|---|---|---|---|---|
| | $\sigma_{\text{before}}$ | $\sigma_{\text{after}}$ | $\sigma_{\text{before}}$ | $\sigma_{\text{after}}$ |
| LOW | 0.6 | 0.3 | 0.0002 | 0.00005 |
| MEDIUM | 1.6 | 0.8 | 0.0004 | 0.0001 |
| HIGH | 2.4 | 1.2 | 100 | 100 |

## D. Experimental Details

We describe some additional details about the experimental setup and the datasets used in our experiments.

### D.1. Routerbench

**Data Split** We use $5\%$ of the RouterBench data (around 2000 samples) to optimize the hyperparameters of cascading, routing, and cascade routing. The remaining $95\%$ is used for evaluation. We use the same data split for all noise levels.

**Noise** In Table 3 we specify the standard deviations of the noise levels on the RouterBench dataset. To put these numbers into context, we note that quality varies between 0 and 1, and the average cost of the smallest models is 0.000073, while the average cost of the largest models is 0.003281. We fit a logistic regression model on this noisy signal to obtain the quality and cost estimates. This simulates the noise in the features that are used to estimate the quality and cost of the models.

**Models** In the evaluated scenarios for three models, we use the models MIXTRAL-8X7B-CHAT, GPT-3.5-TURBO-1106, and GPT-4-1106-PREVIEW. When using five models, we add WIZARDLM-13B-V1.2 and CLAUDE-V2 to the mix. For eleven models, we use all models available in the benchmark.

### D.2. Accurate Quality Estimation

**Data Split** For the SWE-Bench benchmark, we use its verified data split and divide the dataset into training and calibration subsets, with each comprising $50\%$ of the data. For the Minerva Math and LiveCodeBench benchmark, we only include the Algebra portion of Minerva Math to ensure that both benchmarks have a comparable number of samples for evaluation. Similarly, we also perform a $50\%$ split of this dataset into training and calibration sets.

**Evaluation Setting** For the SWE-Bench evaluation, we analyze the performance of 10 models submitted to the benchmark's leaderboard. The logs for these models were obtained from the official SWE-Bench repository[2]. Specifically, we evaluated the following models:

- 20240402_sweagent_claude3opus
- 20241007_nfactorial
- 20240728_sweagent_gpt4o
- 20240620_sweagent_claude3.5sonnet
- 20241016_epam-ai-run-gpt-4o
- 20240824_gru
- 20241106_navie-2-gpt4o-sonnet
- 20240820_epam-ai-run-gpt-4o
- 20241202_agentless-1.5_claude-3.5-sonnet-20241022
- 20241028_agentless-1.5_gpt4o

For each model, we extract the time required to complete a task to measure cost.

---

[2] https://github.com/swe-bench/experiments

For LiveCodeBench and Minerva Math, we evaluate the following models:

- QWEN-2.5-CODER-7B-INSTRUCT
- QWEN-2.5-CODER-1.5B-INSTRUCT
- QWEN-2.5-MATH-7B-INSTRUCT
- QWEN-2.5-MATH-1.5B-INSTRUCT

We conduct experiments using version 5 of the LiveCodeBench benchmark from its official repository. For Minerva Math, we utilize the LM Evaluation Harness (Gao et al., 2024) to ensure consistent and reliable evaluation.

**Cost Estimation**  For SWE-Bench, the cost is defined as the time (in seconds) that a model takes to complete a task. A linear regression model is fitted to predict this cost based on the query length and, when available, the cost of running other models.

For LiveCodeBench and Minerva Math, the cost is calculated as the total number of tokens in both the query and the answer, multiplied by the size of the model (in billions of parameters). Similar to SWE-Bench, a linear model is used to predict the cost based on query length and other models' costs.

**Quality Estimation**  For ex-ante quality estimation in SWE-Bench, we train a logistic regression model that predicts quality based on the query length and a one-hot encoded variable representing the query's source repository. Post-hoc quality estimation leverages the ground-truth quality scores computed during evaluation.

For ex-ante quality estimation in Minerva Math and LiveCodeBench, we include the query length, query source (Minerva Math or LiveCodeBench), and the difficulty level of the problem as defined by the benchmark. Post-hoc quality estimation incorporates additional information, such as whether the parsed answers from different models agree with one another.

### D.3. Poor Quality Estimation

**Data Split**  We split each dataset in each benchmark into a training set and a test set, each comprising $50\%$ of the data. For all datasets except GSM8k, the training set is created by splitting the original test data. In the case of GSM8k, since a separate training set is already available, we use this pre-existing training data, leaving the original test set unchanged. The training set is then further divided, with $50\%$ used for training quality and cost estimators, and the remaining $50\%$ reserved for hyperparameter optimization through validation.

**Evaluation Setting**  We use completion-based evaluation in a one-shot setting for each benchmark. For the classification tasks, we obtain the probability associated with each class ("A", "B", "C", . . . ) from the model directly. For open-form reasoning tasks, we extract the answer by instruction the model to generate a completion that ends with an extractable answer. If the model does not output an answer in the correct format, we perform a best-effort extraction by trying various regex patterns. Details on the prompts and regex patterns used for each benchmark are provided in the code repository.

**Models**  For the LLAMA-3.1 model family, we use the models LLAMA-3.1-8B-INSTRUCT, LLAMA-3.1-70B-INSTRUCT, and LLAMA-3.1-405B-INSTRUCT. For the GEMMA model family, we use the models GEMMA-2B-INSTRUCT, GEMMA-2-9B-INSTRUCT, and GEMMA-2-27B-INSTRUCT. For the MISTRAL model family, we use the models MISTRAL-7B-INSTRUCT-V0.3, MIXTRAL-8X7B-INSTRUCT-V0.1, and MIXTRAL-8X22B-INSTRUCT-V0.1.

**Cost Estimation**  For cost estimation, we first calculate the number of tokens in both the query and the model's response. We then use API-based prices per token for each model to estimate the cost.[3]  In classification, where responses consist of a single token, the cost can be determined before running the model. In open-form reasoning tasks, where response lengths vary, we estimate this length based on responses from previous models in the cascade if the model has not yet been computed. If no model response is available, we estimate the response length using the average from the training data.

---

[3]We used the Together API for all our experiments.

Table 4: AUC scores in % for different strategies on RouterBench in the 0-shot setting with $2\sigma$ confidence intervals.

| | Three Models | | | Five Models | | | Eleven Models | | |
|---|---|---|---|---|---|---|---|---|---|
| | Low | Med | High | Low | Med | High | Low | Med | High |
| Cascade Routing (Ours) | $82.37^{+0.31}_{-0.32}$ | $76.57^{+0.34}_{-0.35}$ | $73.23^{+0.37}_{-0.38}$ | $84.33^{+0.29}_{-0.29}$ | $76.32^{+0.34}_{-0.37}$ | $72.75^{+0.36}_{-0.40}$ | $87.24^{+0.23}_{-0.26}$ | $77.58^{+0.30}_{-0.33}$ | $74.41^{+0.33}_{-0.36}$ |
| − Routing | $2.64^{+0.15}_{-0.16}$ | $1.59^{+0.13}_{-0.15}$ | $1.40^{+0.15}_{-0.17}$ | $3.10^{+0.17}_{-0.15}$ | $1.88^{+0.17}_{-0.16}$ | $1.41^{+0.17}_{-0.17}$ | $4.00^{+0.17}_{-0.21}$ | $2.94^{+0.20}_{-0.21}$ | $1.73^{+0.19}_{-0.19}$ |
| − Cascade (Baseline) | $1.50^{+0.12}_{-0.12}$ | $1.91^{+0.18}_{-0.19}$ | $0.74^{+0.19}_{-0.18}$ | $2.00^{+0.17}_{-0.15}$ | $3.29^{+0.26}_{-0.27}$ | $3.22^{+0.24}_{-0.24}$ | $2.76^{+0.14}_{-0.14}$ | $3.92^{+0.28}_{-0.28}$ | $4.61^{+0.28}_{-0.28}$ |
| − Cascade (Ours) | $1.28^{+0.12}_{-0.11}$ | $0.39^{+0.15}_{-0.14}$ | $0.54^{+0.18}_{-0.17}$ | $1.27^{+0.10}_{-0.11}$ | $1.14^{+0.20}_{-0.21}$ | $2.57^{+0.21}_{-0.26}$ | $2.77^{+0.13}_{-0.13}$ | $2.46^{+0.22}_{-0.24}$ | $4.14^{+0.25}_{-0.27}$ |

Table 5: AUC scores in % for different strategies on RouterBench in the 5-shot setting across model and noise levels with $2\sigma$ confidence intervals. Bold numbers indicate that the confidence interval contains zero.

| | Three Models | | | Five Models | | | Eleven Models | | |
|---|---|---|---|---|---|---|---|---|---|
| | Low | Med | High | Low | Med | High | Low | Med | High |
| Cascade Routing (Ours) $83.79^{+0.3}_{-0.33}$ | $78.85^{+0.29}_{-0.34}$ | $77.1^{+0.32}_{-0.35}$ | $85.5^{+0.26}_{-0.26}$ | $78.77^{+0.32}_{-0.33}$ | $76.74^{+0.34}_{-0.36}$ | $88.78^{+0.22}_{-0.22}$ | $80.89^{+0.28}_{-0.29}$ | $78.03^{+0.3}_{-0.31}$ | |
| − Routing | $2.3^{+0.14}_{-0.13}$ | $1.64^{+0.15}_{-0.15}$ | $1.1^{+0.13}_{-0.14}$ | $3.08^{+0.16}_{-0.14}$ | $1.94^{+0.16}_{-0.14}$ | $1.21^{+0.14}_{-0.15}$ | $3.43^{+0.15}_{-0.17}$ | $3.13^{+0.19}_{-0.2}$ | $1.6^{+0.17}_{-0.16}$ |
| − Cascade (Baseline) | $-0.64^{+0.11}_{-0.1}$ | $0.28^{+0.13}_{-0.14}$ | $0.22^{+0.16}_{-0.16}$ | $1.23^{+0.12}_{-0.12}$ | $2.19^{+0.21}_{-0.21}$ | $2.83^{+0.24}_{-0.24}$ | $1.64^{+0.12}_{-0.13}$ | $2.29^{+0.24}_{-0.24}$ | $3.09^{+0.27}_{-0.26}$ |
| − Cascade (Ours) | $1.02^{+0.1}_{-0.09}$ | $\mathbf{0.09^{+0.11}_{-0.11}}$ | $\mathbf{0.1^{+0.14}_{-0.14}}$ | $1.25^{+0.1}_{-0.09}$ | $1.59^{+0.17}_{-0.17}$ | $2.45^{+0.21}_{-0.21}$ | $2.06^{+0.1}_{-0.1}$ | $2.22^{+0.21}_{-0.19}$ | $2.95^{+0.23}_{-0.24}$ |

**Features Quality Estimates** We specify the exact features used for the logistic regression model that serves as the quality estimator in §5.2. First, we include a one-hot encoding of the various datasets in each benchmark. Furthermore, for classification, we include the probability associated with the highest class and the entropy of the class probabilities if the model has been computed. If several models have been computed, we include both whether they agree on their prediction, and the JS-divergence between their class probabilities. For open-form reasoning, we include the perplexity, number of tokens, and several quantiles of the logits if the model has been computed, in accordance with Gupta et al. (2024). If several models have been computed, we also include whether they agree on their prediction.

We note that we train a separate logistic regression model for each history of computed models, and for each model separately as well. Thus we have one linear model for each combination of a target model $m_i$ and computed models $m_{i_1}, \ldots, m_{i_j}$. All the linear models are trained on the training set included in the benchmark.

# E. Confidence Intervals

To check whether the results obtained by cascade routing are significantly higher than our baselines in Tables 1, 2 and 10, we perform bootstrapping on the samples in the dataset. Specifically, we compute the confidence interval associated with the difference between the AUC scores of cascade routing and the baselines. If this difference is positive and its $2\sigma$ confidence interval does not contain zero, we can conclude that cascade routing is significantly better than the baseline. These confidence intervals are reported in Tables 4–6.

# F. Additional Experiments

## F.1. Ablation Study

We conduct an ablation study to examine the impact of various design choices in cascade routing on performance and runtime. Runtime is a critical factor because the overhead introduced by the strategy must be negligible compared to the time required for model computation. If the strategy adds significant overhead, its performance gains may be offset by the increased runtime. We also include an additional ablation that specifically targets runtime on random data in App. F.2.

To investigate this, we repeat the experiment from §5.1 when using all eleven models, testing different variations of cascade routing. We evaluate a slower variation that omits Lemma 1, thereby requiring more supermodels to be evaluated (SLOW), a greedy variation that only considers supermodels of length $j+1$ at step $j$ (GREEDY), and a version that does not compute the expected value when evaluating supermodel quality, using the quality of the best model instead (NO-EXPECT).

**Results** Table 7 presents the results. As expected, the SLOW variation is almost an order of magnitude slower while achieving similar performance. In contrast, both GREEDY and NO-EXPECT are faster but perform worse in the low- and medium-noise scenarios by 0.5% to 1.3%. Interestingly, there is a much smaller performance gap in the high-noise scenario. This is due to the very low variance in the quality estimates, since the linear model used for quality estimation predicts an almost constant value for each query in this scenario, making the expected value computation less important.

Table 6: AUC scores on the realistic benchmarks with $2\sigma$ confidence intervals. Bold numbers indicate that the confidence interval contains zero.

| | SWE-Bench | | Math+Code | Classification | | | Open-Form | | |
|---|---|---|---|---|---|---|---|---|---|
| | 10 Models | 5 Models | Qwen | Llama | Gemma | Mistral | Llama | Gemma | Mistral |
| Cascade Routing (Ours) | $54.21^{+7.49}_{-7.17}$ | $51.20^{+7.44}_{-7.22}$ | $48.51^{+2.95}_{-2.99}$ | $75.56^{+1.22}_{-1.16}$ | $64.89^{+1.36}_{-1.42}$ | $65.02^{+1.40}_{-1.24}$ | $79.95^{+1.28}_{-1.33}$ | $59.70^{+1.62}_{-1.61}$ | $58.77^{+1.42}_{-1.50}$ |
| — Routing | $13.65^{+4.50}_{-4.32}$ | $11.71^{+4.39}_{-4.14}$ | $1.11^{+0.81}_{-0.80}$ | $0.60^{+0.32}_{-0.34}$ | $\mathbf{0.39^{+0.50}_{-0.47}}$ | $\mathbf{0.08^{+0.12}_{-0.10}}$ | $0.56^{+0.38}_{-0.42}$ | $1.26^{+0.57}_{-0.55}$ | $\mathbf{0.02^{+0.05}_{-0.05}}$ |
| — Cascade (Baseline) | $15.36^{+3.90}_{-3.22}$ | $5.17^{+1.69}_{-1.63}$ | $10.77^{+1.71}_{-1.72}$ | $0.71^{+0.30}_{-0.28}$ | $10.51^{+0.62}_{-0.61}$ | $3.79^{+0.73}_{-0.77}$ | $0.65^{+0.23}_{-0.27}$ | $3.47^{+0.37}_{-0.40}$ | $10.45^{+1.15}_{-1.06}$ |
| — Cascade (Ours) | $\mathbf{0.83^{+1.90}_{-1.50}}$ | $\mathbf{0.11^{+1.05}_{-1.04}}$ | $1.82^{+0.58}_{-0.55}$ | $\mathbf{0.06^{+0.15}_{-0.17}}$ | $2.04^{+0.33}_{-0.31}$ | $1.67^{+0.41}_{-0.42}$ | $0.20^{+0.19}_{-0.19}$ | $2.00^{+0.25}_{-0.25}$ | $3.06^{+0.65}_{-0.63}$ |

Table 7: AUC scores and average runtime for variations of cascade routing on RouterBench when using all eleven models.

| | Low-Noise | | Medium-Noise | | High-Noise | |
|---|---|---|---|---|---|---|
| | AUC (%) | Time (ms) | AUC (%) | Time (ms) | AUC (%) | Time (ms) |
| Cascade Routing | 87.29 | 15.26 | 77.61 | 9.53 | 74.41 | 13.68 |
| Slow | 87.30 | 78.88 | 77.61 | 87.72 | 74.40 | 88.76 |
| Greedy | 85.93 | 1.39 | 77.17 | 1.17 | 74.35 | 0.89 |
| No-Expect | 85.98 | 4.78 | 77.11 | 2.49 | 74.35 | 2.08 |

Furthermore, the GREEDY and NO-EXPECT variants perform very similarly, while GREEDY is about twice as fast as NO-EXPECT. This suggests that one should almost always use the normal variant of cascade routing, and only consider the GREEDY variant if runtime is a critical concern. Neither the SLOW nor the NO-EXPECT variant is recommended, as they either perform worse or are significantly slower than the normal variant.

### F.2. Runtime Analysis

We further analyze the runtime of the four variants of cascade routing presented in App. F.1. Specifically, we perform experiments with random data, scaling the number of models to 80 to evaluate the runtime of all variants. Furthermore, we include a fifth variant of cascade routing in the analysis MAX-DEPTH, which restricts cascade routing to a maximum depth of 3 models. MAX-DEPTH does not reduce performance of cascade routing if the optimal depth is less than or equal to 3 models. However, it does significantly reduce the runtime of cascade routing.

For each number of models, we generate 100 data points, each with random quality and cost estimates associated with each model. For each point, we generate the hyperparameters $\lambda_1, ..., \lambda_k$ and $\gamma$ randomly. We then report the average runtime of the five variants of cascade routing in Fig. 3.

The results show the varying computational complexity of the different variants of cascade routing. SLOW has the highest runtime, and becomes computationally too expensive even when using less than 20 models. In contrast, standard cascade routing has a significantly lower runtime, and is able to handle up to 40 models within a 1 second runtime. Its faster variant, MAX-DEPTH, is able to handle up to 80 models within a 1 second runtime. Furthermore, we now also see a clear difference between NO-EXPECT and GREEDY. While GREEDY remains computationally very cheap even for 80 models, NO-EXPECT has a significantly higher runtime, even obtaining higher runtimes than MAX-DEPTH for 80 models.

Thus, the conclusions from App. F.1 are further supported by the runtime analysis: GREEDY is the most efficient variant of cascade routing, while NORMAL is the most efficient variant that does not compromise performance. MAX-DEPTH is a good choice if the optimal depth is known to be less than or equal to 3 models, as it significantly reduces runtime without compromising performance. Since cascades of more than 3 models are rare, MAX-DEPTH is a good choice in practice.

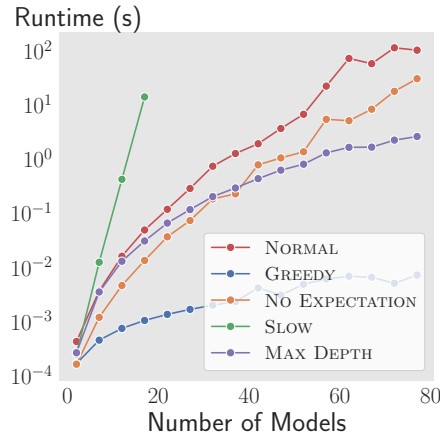

Figure 3: Runtime of cascade routing variants for different numbers of models.

Table 8: Classification AUC values for each benchmark separately for the experiment performed in §5.2.

|  | LLAMA | | | GEMMA | | | MISTRAL | | |
| --- | --- | --- | --- | --- | --- | --- | --- | --- | --- |
|  | MMLU | ARC | MixEval | MMLU | ARC | MixEval | MMLU | ARC | MixEval |
| Linear Interp. | 53.82 | 93.15 | 82.86 | 39.40 | 82.28 | 70.97 | 39.76 | 85.39 | 73.03 |
| Routing | 55.32 | 93.12 | 82.86 | 40.01 | 83.13 | 73.12 | 40.61 | 85.64 | 74.28 |
| Cascade (Baseline) | 54.80 | 94.08 | 84.15 | 36.43 | 77.53 | 66.10 | 36.99 | 83.88 | 72.73 |
| Cascade (Ours) | 55.05 | 94.16 | 84.00 | 37.68 | 79.80 | 70.57 | 37.03 | 86.27 | 74.42 |
| Cascade Routing (Ours) | 55.40 | 93.90 | 83.91 | 39.93 | 83.74 | 73.16 | 40.56 | 86.52 | 74.64 |

Table 9: Open-form AUC values for each benchmark separately for the experiment performed in §5.2.

|  | LLAMA | | GEMMA | | MISTRAL | |
| --- | --- | --- | --- | --- | --- | --- |
|  | MMLU | GSM8k | MMLU | GSM8k | MMLU | GSM8k |
| Linear Interp. | 65.64 | 94.43 | 36.52 | 73.86 | 41.40 | 67.84 |
| Routing | 65.75 | 94.15 | 38.08 | 75.01 | 43.03 | 68.00 |
| Cascade (Baseline) | 66.07 | 95.17 | 35.76 | 68.44 | 38.88 | 60.82 |
| Cascade (Ours) | 66.25 | 94.94 | 38.16 | 71.10 | 40.76 | 64.53 |
| Cascade Routing (Ours) | 66.60 | 94.69 | 40.43 | 75.25 | 42.93 | 68.30 |

## G. Detailed Results

We present benchmark-specific AUC values for the experiment performed in §5.2 in Table 8 for classification and Table 9 for open-form reasoning. In Fig. 4, we show the quality-cost tradeoff curves for several benchmarks. The curves are obtained by varying the cost threshold $\lambda$ and plotting the resulting accuracy and cost values in a curve.

## H. Additional Results

In Table 10 we report the AUC scores for the RouterBench dataset for different noise levels for the five-shot evaluation. Our conclusions presented in §5.1 remain consistent with the results presented in Table 10. However, there is one notable inconsistency: in two of the three low-noise scenarios, our cascading strategy performs worse than the threshold-based baseline cascade. In the scenario with three models, we find its cause can be found in the more difficult optimization surface for the hyperparameters of our cascading strategy. Specifically, our cascading strategy at some point starts to lose quality as cost increases. By simply setting the hyperparameters of the cascading strategy once it starts to lose quality to the ones where it obtained its highest quality, we obtain a quality of $83.35\%$ over the $83.17\%$ of the baseline cascade.

In contrast, for low-noise and eleven models, a similar approach does not yield a better result. Rather, the discrepancy is caused by a small mismatch between the quality estimates of supermodels and the chosen model. While the quality estimate is based on the expected maximum of all models, we restrict the selected model to be the last model that was computed in the cascade. Since the expected maximum is higher than the quality of the last model, this discrepancy can lead to suboptimal decisions. By allowing both the baseline cascade and our cascading strategy to select the model with the highest quality estimate, we find that our cascading strategy once again outperforms the baseline cascade. Note that this slight discrepancy is not relevant for cascade routing, since the extra restriction is not imposed in this setting.

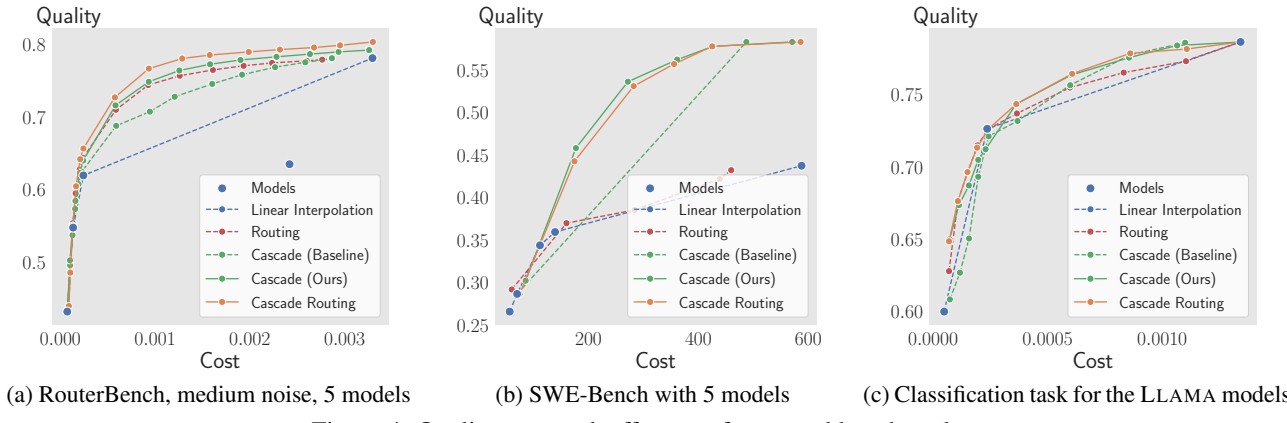

(a) RouterBench, medium noise, 5 models     (b) SWE-Bench with 5 models     (c) Classification task for the LLAMA models

Figure 4: Quality-cost tradeoff curves for several benchmarks.

Table 10: AUC scores in % for different strategies on RouterBench across model and noise levels for five-shot evaluation. Highest numbers are bolded, underlined numbers are within the 95% confidence intervals of the highest number. For a discussion on confidence intervals, we refer to App. E.

| | Three Models | | | Five Models | | | Eleven Models | | |
| --- | --- | --- | --- | --- | --- | --- | --- | --- | --- |
| | Low | Med | High | Low | Med | High | Low | Med | High |
| Linear Interp. | 74.21 | 74.21 | 74.21 | 73.82 | 73.82 | 73.82 | 75.16 | 75.16 | 75.16 |
| Routing | 81.50 | 77.22 | 76.01 | 82.43 | 76.84 | 75.54 | 85.34 | 77.77 | 76.44 |
| Cascade (Baseline) | 83.16 | 78.58 | 76.89 | 84.27 | 76.59 | 73.92 | 87.14 | 78.60 | 74.94 |
| Cascade (Ours) | 82.78 | 78.77 | 77.01 | 84.26 | 77.19 | 74.30 | 86.72 | 78.67 | 75.08 |
| Cascade Routing (Ours) | **83.80** | **78.86** | **77.11** | **85.50** | **78.78** | **76.75** | **88.78** | **80.90** | **78.04** |

Table 11: AUC scores on several benchmarks for the MISTRAL model family. Highest numbers are bolded, underlined numbers are within the 95% confidence intervals of the highest number. For confidence intervals, see App. E.

| | Classification | Open-Form |
| --- | --- | --- |
| Linear Interp. | 63.39 | 53.86 |
| Routing | 64.89 | 58.71 |
| Cascade (Baseline) | 61.20 | 48.29 |
| Cascade (Ours) | 63.31 | 55.51 |
| Cascade Routing (Ours) | **64.97** | **58.73** |

