# OpenReview forum: "A Unified Approach to Routing and Cascading for LLMs"
_ICML.cc/2025/Conference — ICML 2025 poster_

### Official Review · Reviewer_abFg · 2025-03-08

**Overall Recommendation:** 4

**Summary:**

The paper proposes a unified framework termed "cascade routing" that integrates routing and cascading strategies to optimize the selection of large language models (LLMs) based on a cost-performance tradeoff. It gives a theoretically grounded method using linear optimization to derive optimal routing and cascading strategies, supported by proofs of optimality. The authors provide mathematical formulations and theorems, and present experimental results across multiple benchmarks such as RouterBench and SWE-Bench. These experiments demonstrate that cascade routing outperforms routing and cascading baselines.

## update after rebuttal
Thank the authors for their response, which has clarified most of my concerns. Overall, I find the paper's contributions to the theoretical framework—particularly in unifying routing and cascading via the concept of "supermodel"—to be quite novel. I would personally vote for acceptance.

**Claims And Evidence:**

The claims are overall well supported. See weakness and questions below for details.

**Essential References Not Discussed:**

The paper would benefit from citing previous works on dynamic thresholds for cascading [1,2]. These could provide a more competitive baseline than the simplistic cascades currently evaluated.

[1] Jitkrittum, Wittawat, et al. "When does confidence-based cascade deferral suffice?." Advances in Neural Information Processing Systems 36 (2023): 9891-9906.
[2] Nie, Lunyiu, et al. "Online cascade learning for efficient inference over streams." Proceedings of the 41st International Conference on Machine Learning. 2024.

**Experimental Designs Or Analyses:**

The evaluation is compromised by synthetic noise in RouterBench, which may not mirror real-world conditions. Also, incorporating stronger baselines, such as adaptive thresholding methods (See missing references below), would offer a more meaningful comparison.
Experiments also show minimal gains under poor quality estimation, underscoring the framework’s sensitivity to this factor. The analysis lacks depth regarding robustness to estimation errors, limiting its interpretability.

**Methods And Evaluation Criteria:**

The chosen benchmarks are commonly-used benchmarks and are relevant to the task of LLM model selection. The use of AUC as a metric is appropriate for evaluating cost-performance trade-offs. See weakness and questions below for details.

**Other Comments Or Suggestions:**

Typo: line 961, "we extract the answer by instruction the model", instruction" should be "instructing"
See below for other suggestions.

**Other Strengths And Weaknesses:**

Strengths:
I really like the idea of framing cascading as routing among "supermodels." It’s a clever way to unify routing and cascading into a single theoretical framework that feels both fresh and neat. The convex combination approach is novel and provides new insights in achieving better cost-performance trade-offs. The theoretical claims are supported by detailed mathematical derivations and proofs provided in the appendices, and the experimental results on multiple benchmarks provide convincing empirical evidence.

Weaknesses:
1. In the analysis, restricting cascading to be deterministic is oversimplifying this line of approaches. There are many works [1-3] that dynamically adapt the confidence thresholds in cascades by a separate post-hoc confidence calibration model. I would suggest comparing with stronger baselines in the experiments.
2. While the proofs are mostly sound, certain ambiguities in the proof make it unclear to the readers.
- The assumption $|\Lambda| < \infty$ (finite points where tradeoffs equalize) is justified by finite precision in practice but limits theoretical generality. In continuous spaces, $ \Lambda $ could be infinite.
- The proof assumes a convex combination exists for any $ B $, but if $ S_{\lambda^*} $ contains multiple strategies with varying costs, selecting $ \gamma $ to hit $ B $ precisely is not detailed. The proof states it "must exist," but the mechanism is opaque.
- In Appendix B, a single $ \gamma $ across steps simplifies the problem but lacks justification beyond the global budget constraint. Step-specific $ \gamma_j $ might better reflect varying tradeoffs.
- For the proof regarding the equivalence of Thresholding and Cascading, I found the conditions restrictive—constant costs and query-independent quality estimates rarely hold in practice, and the supermodel quality assumption feels contrived.
- Regarding the cascade routing approach, the exponential growth of supermodel combinations is acknowledged but unresolved theoretically. Computational complexity is highlighted in Appendix F and mitigated by restricting max. depth, but the proof assumes full evaluation, which is infeasible for large $ k$.
3. In practice, quality estimation—whether ex-ante or post-hoc — is not free and must come with a time/cost that can significantly influence the evaluation of the proposed cascade routing framework. However, the paper assumes that quality estimates $\hat{q}_i(x)$ and cost estimates $hat{c}_i(x)$ for each model and query are readily available. If the time/cost of quality estimation is high, it could offset the advantages of the framework, especially in applications where efficiency is important.

[1] Jitkrittum, Wittawat, et al. "When does confidence-based cascade deferral suffice?." Advances in Neural Information Processing Systems 36 (2023): 9891-9906.
[2] Nie, Lunyiu, et al. "Online cascade learning for efficient inference over streams." Proceedings of the 41st International Conference on Machine Learning. 2024.
[3] Enomoro, Shohei, and Takeharu Eda. "Learning to cascade: Confidence calibration for improving the accuracy and computational cost of cascade inference systems." Proceedings of the AAAI Conference on Artificial Intelligence. Vol. 35. No. 8. 2021.

**Questions For Authors:**

1. It's unclear to me in the experiments how 𝛾 is determined in practice.
2. In cascading, would step-specific $ \gamma_j $ improve performance by reflecting varying tradeoffs?
3. Quality estimation isn’t free in practice. Is this factored into the framework when evaluating the AUC score?
4. The experimental results look good, but the lack of stronger baselines (e.g., adaptive cascades) —please consider addressing these in revisions.
5. Adding more illustrative examples in the paper to aid readers in understanding the convex combination approach and the dynamic routing decisions in cascade routing.
6. Does the notation $ M_{1:j-1}, \ldots, M_{1:k} $ in cascade routing imply a fixed sequence, or is it just a label? Please clarify, as it seems to conflict with the flexibility of cascade routing.
6. The "Model Order" paragraph in Sec 4 is a bit confusing -- if models are sorted by cost within supermodels, is the model invocation order in cascade routing fully flexible? If so, can you provide an example of a non-sequential order to illustrate this?

**Relation To Broader Scientific Literature:**

Overall the paper situates itself within the broader field of model selection for LLMs, building on established concepts while introducing a novel unification of LLM routing and cascading.

**Theoretical Claims:**

I looked at the proofs in Appendix but did not check the details. There are some fuzzy spots—like how they pick certain parameters and some limiting assumptions—that make the paper a bit unclear. See weakness and questions below for details.

---

> ### Author Rebuttal · Authors · 2025-03-31
>
> We thank the reviewer for their review. We are happy to hear that they found our claims well supported, our theoretical reformulation fresh and neat, and our experiments convincing. Below, we address their questions.
>
> **Can you clarify how $\gamma$ is determined and whether step-specific $\gamma_j$ factors might improve performance?**
> First, we highlight that this parameter is generally of minor significance, becoming relevant only when $s_{\min}$ and $s_{\max}$ differ. This scenario is limited to cases where two or more models achieve exactly identical trade-offs, which occurs rarely in practice.
>
> The value of $\gamma$ is determined according to the procedure outlined in lines 701-702: we first estimate the costs associated with $s_{\min}$ and $s_{\max}$ using training data, and then select $\gamma$ to precisely satisfy the budget constraint $B$ by solving $B = \gamma B^- + (1 - \gamma) B^+$. Since this equation is always invertible, as $B \in [B^-, B^+]$, $\gamma$ exists.  Due to linearity in cost, the resulting optimal routing algorithm, $\gamma s_{\min} + (1 - \gamma) s_{\max}$, matches the budget exactly.
>
> Since the only purpose of $\gamma$ in the proof is to ensure that the cost of the final strategy exactly matches $B$, using step-specific $\gamma_j$ will not significantly alter the strategy and would yield equivalent theoretical quality.
>
> **Is the cost of quality estimation factored in when evaluating the AUC score?**
> In all our experiments, the cost of quality estimation was sufficiently negligible to exclude it from the AUC score. Specifically, we used small linear models based on model confidence or categorical features. Running these linear models essentially does not cost anything. The reviewer correctly points out that cheap quality estimates are essential. However, we do want to point out that all current algorithms suffer from this limitation.
>
> **Did you include baselines based on adaptive thresholds?**
> Yes, as the presented baseline cascade does allow for adaptive thresholds. All works cited by the reviewer can be reformulated under the thresholding mechanism presented in Corollary 1. Specifically, by appropriately selecting the estimator $q_{j-1}^{(j)}(x)$, our formulation becomes equivalent. For example, for [1], one can set $q_{j-1}^{(j)}(x) = \eta_{h^2}(x) - \eta_{h^1}(x)$, thereby exactly recovering their scheme. We will clarify this in the revised paper.
>
> We also agree with the reviewer’s observation regarding the restrictive nature of the optimality conditions presented in Corollary 1. We stress that these strict conditions apply exclusively to the optimality of the baseline thresholding mechanism, and not to our newly introduced cascading approach.
>
> **Can you add some illustrative examples of cascade routing?**
> Yes, reviewer bNjc similarly suggested that adding an algorithm block for cascade routing would enhance clarity. We will incorporate both an algorithmic description and an illustrative example into the paper.
>
> **Does cascade routing only look at the supermodels $M_{1:j-1}, \dots, M_{1:k}$?**
> No, the given subset of supermodels is only required during cascading. As stated in lines 260-261, we specifically remove this requirement from cascade routing, thereby allowing consideration of all possible supermodels.
>
> **Can you clarify what you mean in the model ordering paragraph?**
> This paragraph serves only to identify the optimal initial model to execute within the chosen supermodel. To illustrate, suppose we label models from least to most expensive as $m_1, \dots, m_k$. If the cascade routing algorithm initially selects the optimal supermodel as {$m_3, m_5$}, the model $m_3$ will be executed first. Subsequently, after updating quality and cost estimates, the algorithm may identify a new optimal supermodel, for instance, {$m_1, m_2, m_3$}. Since $m_3$ has already been executed, it remains part of the optimal set, and thus $m_1$ will be run next. We will clarify this in the paper.
>
> **Could you add a discussion regarding robustness of the method to estimation errors?**
> Yes, we provide an empirical discussion of the robustness of all methods by discussing ex-ante and post-hoc quality estimation and showing performance for different estimation errors in Figure 2. These discussions say where both cascading and routing are most helpful, and in which cases cascade routing outperforms them most. However, we were not able to prove any interesting results regarding the effect of estimation errors on performance as this is a very difficult problem.
>
> **Does restricting the maximum depth of cascade routing significantly affect performance?**
> The reviewer accurately notes that our theoretical proof imposes no explicit restriction on the number of models executed. However, we wish to emphasize that, in practical settings, cascade routing very rarely selects more than five models. Thus, enforcing a maximum cascade depth has negligible practical impact.

---

> > ### Comment · Reviewer_abFg · 2025-04-04
> >
> > I appreciate the authors for their detailed response, which addressed most of my questions and concerns. Please ensure you incorporate these clarifications and discussions into the revision, and include the missing references [1, 2] for better positioning of the work.
> >
> > [1] Jitkrittum, Wittawat, et al. "When does confidence-based cascade deferral suffice?." Advances in Neural Information Processing Systems 36 (2023): 9891-9906.
> >
> > [2] Nie, Lunyiu, et al. "Online cascade learning for efficient inference over streams." Proceedings of the 41st International Conference on Machine Learning. 2024.

---

### Official Review · Reviewer_wKyv · 2025-03-13

**Overall Recommendation:** 3

**Summary:**

This paper studies how to use multiple LLMs to improve overall performance under budget constraints. The key idea is to combine two popular approaches, model routing and model cascade. The authors start with analyzing model routing, and then generalizes this analysis to multiple rounds of model routing, which the authors term as cascade routing. Experiments with real-world datasets are performed and analyzed.

**Claims And Evidence:**

- The overall idea of combing cascade and routing seems reasonable, although a little incremental.

- Optimal routing strategy: A main claim is that prior work cannot express the optimal solution derived by Theorem 1. This is based on the assumption that "it can occur that several models achieve the same optimal cost-quality tradeoff for a given query". But isn't this a pathological case? As long as the cost is different for all models, then the cost-quality tradeoff will be different for all models which can answer the user question correctly. Among all models that answer it correctly, the optimal solution is simply the one with the lowest cost. In other words, given the limited scope, I am unsure if the complex analysis is worth it.

**Essential References Not Discussed:**

NA

**Experimental Designs Or Analyses:**

The experiments only report AUC by varying the cost from the cheapest to the most expensive model's price. However, the cost of a cascade approach may be higher than the most expensive model since it may call multiple models for a given question. Following (Hu 2024), I would also be curious to see the performance-cost tradeoff curves.

**Methods And Evaluation Criteria:**

The method sounds reasonable, but the evaluation metric is a bit limited. See my comments later.

**Other Comments Or Suggestions:**

NA

**Other Strengths And Weaknesses:**

NA

**Questions For Authors:**

1. In line 216, why is the expectation of the maximum but not just q_i?

2. How does the proposed cascade routing perform with more recent models, such as GPT-4o, Claude 3.5 Sonnet, and Gemini 1.5 Pro?

3. How does the latency of cascade routing compare with the baselines?

4. What is the trade-off curve (acc/AUC/exact match vs cost) for the proposed cascade routing?

5. How do the authors predict the model quality on ROUTERBENCH?

**Relation To Broader Scientific Literature:**

NA

**Theoretical Claims:**

No, I only scanned the analysis.

---

> ### Author Rebuttal · Authors · 2025-03-31
>
> We thank the reviewer for their detailed review. Below, we address their remaining questions.
>
> **Why is the expectation of the maximum but not just $q_i$?**
> There are two primary reasons for using the expectation of the maximum. First, while the model $m_i$ is often stronger than the model $m_{i-1}$, this does not hold universally. For example, our experiments for the LiveCodeBench and Minerva benchmarks include four models: a small and large math model and a small and large code model. The math models perform better on mathematical queries, while the code models perform better on code queries. It is impossible to rank these four models such that each subsequent model $m_i$ is always better than $m_{i-1}$ for all queries.
>
> The second reason is more technical: the suggested change would imply that the quality of the supermodel is equal to the quality of its best-performing model. Yet, the supermodel has higher costs compared to using the best individual model alone. Thus, the cost-quality tradeoff of such a supermodel would be strictly worse, and cascading routing would therefore never select a supermodel consisting of more than one model. This would reduce it to a routing strategy.
>
> **Did you include the latest state-of-the-art models?**
> Yes, our experiments on SWE-Bench employ state-of-the-art agents, primarily based on advanced models such as GPT-4o and Claude-3.5-sonnet. Additionally, our experiments on LiveCodeBench and Minerva use the state-of-the-art open-source Qwen-2.5 model family. Although Llama-3 is no longer considered the absolute state-of-the-art due to recent advances, it remains a robust and widely used model family. Thus, our experiments confirm that cascade routing is effective when applied to current state-of-the-art models.
>
> **How does the latency of cascade routing compare with the baselines?**
> Generally, the latency of cascade routing falls between pure routing and pure cascading strategies, as it can selectively execute multiple models (increases latency) or skip models (reduces latency). Moreover, for specific applications, latency costs can be explicitly incorporated into the cascade routing cost function, enabling direct optimization for latency. For instance, the cost in our SWE-Bench benchmark is measured as the total time in seconds it takes an agent to complete the task. As can be seen there, cascade routing is much better than routing for the same latency. This is because cascade routing can decide to run cheaper models first before executing the larger model that has higher latency. We will clarify this point in our revised paper.
>
> **Can you clarify how you estimated quality on the RouterBench benchmark?**
> As explained in lines 297-308, we use random noise estimators for RouterBench. Specifically, we add normal noise of varying strength to the ground-truth quality and cost. This enables us to simulate real-world scenarios with varying fidelities of quality and cost estimators. For instance, high noise would simulate scenarios with poor estimators. This enables us to systematically evaluate our algorithm across varied conditions. Specifically, Figure 2 illustrates the extent of improvement provided by cascade routing over baseline methods under different noise scenarios. This choice is very similar to the choice by the authors of RouterBench, who use a slightly different form of this random noise.
>
> **Is one of your main claims that the routing strategy is better than prior work?**
> No, we have been careful throughout the paper not to claim that our proposed routing algorithm is a major improvement over existing methods. For instance, in lines 77-79, we explicitly state that the algorithm introduced in Section 2 closely resembles previous work. Our primary contribution in Section 2 is providing a clean and rigorous proof of optimality for this approach. Our significant algorithmic advancements over prior methods appear in Sections 3 and 4. Specifically, the algorithms for cascading and cascade routing represent substantially novel contributions. Thus, the analysis presented in Section 2 primarily serves as a foundation for introducing and validating these new algorithms in Sections 3 and 4: without this analysis, demonstrating their optimality would not be possible.
>
> **Can you provide performance-cost tradeoff curves?**
> Yes, we will incorporate these curves in our revision. Generally, they closely resemble those presented in Hu (2024), confirming our averaged results and clearly demonstrating that cascade routing consistently outperforms other approaches.

---

### Official Review · Reviewer_4YAD · 2025-03-13

**Overall Recommendation:** 3

**Summary:**

Existing routing and cascading serve as two distinct strategies for LLMs. This work provides a theoretical analysis of the optimality of existing routing strategies and further proposes cascade routing that integrates both routing and cascading as a theoretically optimal strategy. Cascade routing frames the problem as a linear optimization problem by maximizing output quality within a limited computation budget. In the experiments, cascade routing shows improvements over individual routing or cascading approaches.

## update after rebuttal
I thank the author for the rebuttal, which mostly addressed my concerns. Therefore, I have increased my score from 2 to 3. However, I still find the quality estimation metric studied in this work questionable and limited, which I consider as the core for improving routing and cascading.

**Claims And Evidence:**

There are no particular claims made in the paper.

**Essential References Not Discussed:**

[1] Damani, M., Shenfeld, I., Peng, A., Bobu, A., & Andreas, J. (2024). Learning How Hard to Think: Input-Adaptive Allocation of LM Computation (No. arXiv:2410.04707). arXiv. https://doi.org/10.48550/arXiv.2410.04707 [ICLR 2025]

**Experimental Designs Or Analyses:**

The experiments are conducted on RouterBench and Practical Benchmarks that are reasonable.

**Methods And Evaluation Criteria:**

The proposed method, cascade routing, makes sense by reframing the optimization problem as a linear optimization problem with a budget constraint. The evaluation set is fair. However, quality estimation is questionable.

**Other Comments Or Suggestions:**

It would be beneficial if the author could include the training cost, the inference latency, and the inference cost compared to individual baselines in the main body of the work, together with the main table.

**Other Strengths And Weaknesses:**

Strength:
1. This work provides a good optimality analysis of the individual routing and cascading strategies.

2. The discussion on the ex-ante quality estimation and post-hoc quality estimation is great.

3. The proposed cascade routing shows improvement over individual routing and cascading strategies.

Weakness
1. This work is highly reliant on an accurate quality estimation. However, in many real scenarios, the quality is often difficult to estimate, and simply using uncertainty is not convincing enough to show the optimality of the proposed approach, since LLMs are exposed to miscalibrated predictions.

2. The baselines are relatively weak. There are a few competitive routing baselines but have not been included and compared in this work. [1]

3. The increased inference cost has not been discussed explicitly in the work. In addition, the requirement of a training set, and the actual stability with respect to the size of the training set are not discussed.

**Questions For Authors:**

1. Instead of considering the optimal strategy as a linear optimization problem, does it make more sense to frame it as a bi-level optimization, minimizing the cost while maximizing the performance, such that leading to a better Pareto-front?

**Relation To Broader Scientific Literature:**

The routing and cascading is important for efficient LLM serving and deployments.

**Theoretical Claims:**

This work formulates the optimal routing strategy and optimal cascading strategy in Theorem 1 and 2, respectively. It argues that cascade routing can also serve as the optimal strategy. The formulation is correct. However, the assumption is built on top of a perfect estimation of the quality and cost of the query. Therefore, the final strategy may not be optimal.

---

> ### Author Rebuttal · Authors · 2025-03-31
>
> We thank the reviewer for their review. We are pleased that they appreciated our optimality analysis of the strategies, our comprehensive discussion of quality estimation, and the demonstrated improvement of our cascade algorithms over baselines. Below, we address their remaining questions.
>
> **Could you formulate cascade routing as a bi-level optimization problem?**
> Yes, one could formulate cascade routing as a bi-level optimization problem. However, such a bi-level formulation would yield solutions identical to those derived through our current linear optimization approach. To argue why bi-level optimization would not lead to further improvements, suppose we have two strategies $s_1$ and $s_2$ that achieve the same quality, but $s_1$ is cheaper than $s_2$ (both below the allocated cost budget $B$). Since our algorithm optimizes the cost-quality tradeoff, it would always select $s_1$ as the final strategy, and not $s_2$. Therefore, bi-level optimization would be completely equivalent to our current approach.
>
> **Why have you not included the baseline Learning How Hard to Think: Input-Adaptive Allocation of LM Computation?**
> The strategy presented in this paper is equivalent to our baseline routing strategy instantiated with an application-specific quality estimator. Specifically, the quality estimator is the inverse of the difficulty predictor introduced in the cited paper. Thus, the innovations from the cited work can also enhance cascade routing performance when used for quality estimation.
> Our experimental section already includes multiple diverse quality estimators, and due to constraints on space and time, we could not explore all possible estimators. Importantly, our primary goal is to demonstrate that cascade routing consistently outperforms other strategies across different instantiations of quality and cost estimators. We will clarify this relationship explicitly in the revised paper, illustrating how the referenced work can be seen as an instantiation of our routing strategy.
>
> **Does the optimality proof require good quality estimation?**
> No, we want to stress that the optimality proof holds even in the presence of bad quality estimators. If all that is available are bad quality estimators, no algorithm could outperform cascade routing and obtain better performance. However, better quality estimators could increase the predictive power of algorithms like ours. Thus, our algorithm achieves optimality given current estimation techniques.
> We further emphasize that even with suboptimal quality estimators, cascade routing consistently outperforms baseline strategies. Given that most research in this area focuses precisely on improving quality estimation, these innovations can be directly integrated into cascade routing to further enhance performance.
>
> **What about the increased inference and latency costs?**
> Cascade routing does not inherently increase inference costs. On the contrary, our results indicate that, under identical budget constraints, cascade routing achieves higher accuracy.
> More importantly, latency costs can be explicitly incorporated into the cascade routing cost function, enabling direct optimization for latency. For instance, the cost in our SWE-Bench benchmark is measured as the total time in seconds it takes an agent to complete the task. As can be seen there, cascade routing is much better than routing for the same latency. This is because cascade routing can decide to run cheaper models first before executing the larger model that has higher latency. We will clarify this point in our revised paper.
>
> **What are the requirements for the training data?**
> The amount of training data required for fitting cascade routing parameters is generally modest, typically consisting of only a few hundred samples. The associated training overhead is minimal since the procedure involves fitting merely $k+1$ parameters, making it easily executable on standard hardware within a short duration. We further found during our experimentation that the impact of the training data on output quality remains minor, provided it reasonably resembles the benchmark data.
>
> **Could you provide supplementary materials?**
> Contrary to the reviewer's observation, we confirm that our code was included as supplementary material. Additionally, we have provided extensive supplemental details in the appendix.

---

### Official Review · Reviewer_bNjc · 2025-03-19

**Overall Recommendation:** 3

**Summary:**

The paper proposes to combine cascading and routing, two common approaches for inference with multiple LLMs. The authors formulate each as an optimization problem and then solve it to derive optimal routing and cascading approaches. Finally, they propose "cascade routing" which is a generalized optimization formulation that can be solved to derive a combination of cascading and routing with multiple models where at each level of the cascade, one of the models is selected for inference. Evaluation results show that the approach outperforms approaches that do only cascading or routing.

## update after rebuttal

While the authors have responded to all of my questions with adequate clarification, and I feel the paper makes an interesting contribution, I still feel that the writing is very dense, and the exact algorithm is hard to follow. I am not sure how much this can be rectified at the camera-ready stage and therefore I am keeping my score at 3.

**Claims And Evidence:**

Yes

**Essential References Not Discussed:**

N/A

**Experimental Designs Or Analyses:**

I have checked the experiments in the main paper, and I do not have any issues with their validity.

**Methods And Evaluation Criteria:**

Yes

**Other Comments Or Suggestions:**

I would strongly recommend adding an algorithm block in the main paper or appendix to explain the cascade routing approach clearly.

**Other Strengths And Weaknesses:**

Some of the details are not clear (see questions below)

**Questions For Authors:**

1. The role of the superscript (j) in Theorem 2 is not clear.

2. The role of variance in estimating the expected max quality of the models in the cascade is not clear to me.

3. Why do you only consider the extremes of cost in defining in $s_\min$ and $s_\max$? Can't we similarly define strategies for the extremes of quality?

4. Why, in Lemma 1, can all supermodels containing all models in $M$ be pruned from the search space if one model $m \in M$ negatively impacts the quality cost tradeoff? Shouldn't we just prune $m$ and not all models in $M$?

**Relation To Broader Scientific Literature:**

Strengths:

1. The novel cascading approach proposed by the paper can improve over existing cascading approaches and provide a better understanding of LLM cascades

2. The combination of cascading and routing has the potential to combine the best of both worlds

Weaknesses:

1. The optimization problem requires careful selection of hyperpaparameters and it is not clear how bad an incorrect choice would be

2. The explanation of Cascade Routing in Section 4 is quite dense. Specifically, it is not clear how the approach selects the next supermodel at each step or what the exit criteria is.

**Theoretical Claims:**

I checked the proofs at a high level, and they appear correct to me.

---

> ### Author Rebuttal · Authors · 2025-03-31
>
> We thank the reviewer for their review. We are happy to hear that they found our experiments valid, our approach novel, and that we provide a better understanding of cascading. Below, we address their remaining questions.
>
> **What is the role of the superscript (j) in Theorem 2?**
> The superscript $(j)$ indicates that the quality estimator can update each time a model is executed. For instance, $q^{(0)}$ refers to the quality estimator before executing any model. For example, $q^{(0)}$ could predict quality based on the average accuracy across a training dataset. After model execution, $q^{(1)}$ can use the newly obtained data to refine quality estimates. For example, if the executed model generates long reasoning, it might suggest a more challenging question, requiring the estimator to reduce the accuracy predictions.
>
> **What is the role of the variance for quality estimation?**
> Quality estimators are often not exact, meaning there is an uncertainty in their prediction. Modeling this uncertainty enables us to find more optimal solutions. For instance, variance allows us to capture low-probability events where the output of a model is “surprisingly” bad given the quality estimate. The linearity of the routing problem ensures that the variance does not influence the optimal solution. In contrast, the $\max$ operator used for quality estimation in supermodels for cascading is not linear, and therefore gives a different solution when incorporating variance.
>
> More explicitly, the equation  $\mathbb{E}(\max(\hat{q}_1(x), …, \hat{q}_k(x)))$ would simplify to $\max(\hat{q}_1(x), …, \hat{q}_k(x))$. Thus, the quality of the supermodel would be equal to the quality of its best-performing model. Yet, the supermodel has higher costs compared to using the best individual model alone. Thus, the cost-quality tradeoff of such a supermodel would be strictly worse, and cascading routing would therefore never select a supermodel consisting of more than one model. However, by incorporating variance, the quality of the supermodel improves. Specifically, unexpectedly poor predictions from the best individual model can be compensated by better predictions from other models. This aggregation effect results in a better cost-quality tradeoff.
>
> **How are hyperparameters selected?**
> Almost all parameters are determined automatically. As shown in Equation (2) on Line 192,  $\lambda_1, \dots, \lambda_k, \gamma$ are all determined using an automated tool and a small calibration set. The only hyperparameter that requires manual setting, is the cost budget $B$. This hyperparameter is very interpretable, and cannot be set automatically as it depends on the specific use-case and budget of the user.
>
> **Can you define the extremes of quality strategies instead of cost strategies?**
> Both approaches are equivalent and would lead to the same optimal solution! Selecting the cheapest model that achieves the same optimal tradeoff is equivalent to selecting the one with lowest quality. Indeed, if $q_1 - \lambda c_1 = q_2 - \lambda c_2$ and $c_1 < c_2$, then $q_1 < q_2$. Thus $s^\lambda_\min$ and $s^\lambda_\max$  are the extremes of both cost and quality.
>
> **How exactly are you pruning the search space?**
> We are pruning supermodels from the set of all supermodels, not models from the set of all models. If a supermodel $M_1$ satisfies the pruning conditions in Lemma 1, we prune all supermodels $M_2$ that are a superset of $M_1$, i.e., $M_1 \subseteq M_2$. To illustrate, suppose we have four models, and the presence of $m_2$ negatively impacts the cost-quality tradeoff of the supermodel {$m_1, m_2$}. Therefore, all supermodels containing both $m_1$ and $m_2$ are pruned. Thus, instead of evaluating all 16 supermodels, we would exclude {$m_1, m_2, m_3, m_4$}, {$m_1, m_2, m_3$}, {$m_1, m_2, m_4$}, and {$m_1, m_2$}. We will clarify this.
>
> **Can you add an algorithm block explaining cascade routing?**
> Yes, we will incorporate the recommended algorithm block. Additionally, we will provide an illustrative example.
>
> **What is the exit criterion?**
> The algorithm stops if the current supermodel is deemed to be the most optimal one. For example, suppose two steps into our cascade routing process we have executed models $m_1$ and $m_4$. If the algorithm predicts the supermodel {$m_1, m_4$} as optimal, it stops. Conversely, if it selects {$m_1, m_4, m_5$} as optimal, the process continues by executing $m_5$.
>
> **How does your approach select the next supermodel?**
> Cascade routing computes the quality-cost tradeoff for all possible supermodels and selects the supermodel offering the best tradeoff. Continuing with the example above, “all possible supermodels” refers to every supermodel containing $m_1$ and $m_4$, as these have already been executed. Among these, the algorithm selects the supermodel with the best cost-quality tradeoff. If the best supermodel is {$m_1, m_4, m_5$}, cascade routing proceeds by executing $m_5$.

---

> > ### Comment · Reviewer_bNjc · 2025-04-09
> >
> > Thank you for answering my questions in the rebuttal. As I had already recommended accepting the paper, I will keep my score unchanged.

---

### Decision · Program_Chairs · 2025-05-01

**Decision:**

Accept (poster)

**Comment:**

Cascading and routing are two common strategies to reduce LLM inference costs: both these strategies leverage a pool of LLMs, and select the most suitable LLM for a given query. The authors derive the optimal strategy for routing and cascading, and provide a new strategy for cascading that has better optimality guarantees than prior approaches. They also propose a hybrid strategy that combines cascading and routing, and experimentally demonstrate its efficacy over pure cascading and pure routing baselines.

The reviewers generally appreciate the unified theoretical analysis and find the proposed cascade routing strategy to be a useful contribution. However, there were concerns about the writing being unclear in some places, the method relying on the availability of good quality estimators, the need for heavy hyper-parameter optimization, and the baselines being relatively weak. Many of these concerns were addressed during the rebuttal stage. We trust the authors will incorporate all the changes promised, including providing clarifications about the baselines, and providing an algorithm block and an illustrative example.

Based on my own reading of the paper, I really liked the formulation of cascading and routing under a unified framework. I have the following comments that I strongly encourage the authors to address in their camera-ready paper:
- **Routing to supermodels**: In Section 3, the authors view cascading as a sequential routing problem, where at each step $j$, they route to one of the supermodels $M_{1:j-1}, \ldots,  M_{1:k}$. What would happen in this formulation when we decide to route to supermodel $M_{1:i}$, where $i > j$? Would you simply choose to call the next model $m_j$, or directly call model $m_i$? I guess it is the former, but this wasn't clear from the discussion after Definition 4. A similar question arises in Section 4. With a cascade router, what does it mean to route to supermodel $M$ at step $j-1$: would this result in invoking the smallest model in $M$ or *all* models in $M$?

- **Randomized routing:** In your optimality analysis, you allow for a randomized routing strategy, providing greater flexibility compared to prior work. However, it is important that you discuss the feasibility/difficulty of employing a randomized router in practice. Furthermore, even in a setting where we are allowed only deterministic routing decisions, it might be worth mentioning that the optimal router can be shown to be of the form $s^\lambda(x) = {\rm argmax}_i ~q_i(x) - \lambda\cdot c_i(x)$ (i.e., a single deterministic router) by making *continuity assumptions on the data distribution* (via a Neyman-Pearson like analysis). For e.g. Theorem 6 in [2] derives the optimal deterministic classifier for the case of two models; Proposition 1 in [3] derives this for the multi-model case.

- **Quality estimator details:** You mention that in your real-world experiments, you use a linear model on log-probability features. It might be worth clarifying if you use the log-probabilities from *all* models invoked at the current step. More generally, it might be worth elaborating on the construction of the quality estimator in the main text (perhaps move some details from the appendix), as this is a crucial component in your proposal.

- **Relationship to [2]:** The paper by Trapeznikov et al. (2013) [1] also present some optimality results for sequential routing, and the results have some similarity to your results for cascading (see e.g, the optimal solution in Theorem 1 in their paper). Might be worth having a discussion on the connections to this paper.


[1] Supervised Sequential Classification Under Budget Constraints. AISTATS 2013.

[2] Learning to Reject Meets Long-tail Learning. ICLR 2024.

[3] Universal Model Routing for Efficient LLM Inference, arXiv:2502.08773 , 2025.